# O-GlcNAc transferase inhibits visceral fat lipolysis and promotes diet-induced obesity

Yunfan Yang[1], Minnie Fu[1], Min-Dian Li[1,2], Kaisi Zhang[1,2], Bichen Zhang[1,2], Simeng Wang[1], Yuyang Liu[1], Weiming Ni[1], Qunxiang Ong[1], Jia Mi[1] & Xiaoyong Yang [1,2]*

Excessive visceral fat accumulation is a primary risk factor for metabolically unhealthy obesity and related diseases. The visceral fat is highly susceptible to the availability of external nutrients. Nutrient flux into the hexosamine biosynthetic pathway leads to protein post-translational modification by O-linked β-N-acetylglucosamine (O-GlcNAc) moieties. O-GlcNAc transferase (OGT) is responsible for the addition of GlcNAc moieties to target proteins. Here, we report that inducible deletion of adipose OGT causes a rapid visceral fat loss by specifically promoting lipolysis in visceral fat. Mechanistically, visceral fat maintains a high level of O-GlcNAcylation during fasting. Loss of OGT decreases O-GlcNAcylation of lipid droplet-associated perilipin 1 (PLIN1), which leads to elevated PLIN1 phosphorylation and enhanced lipolysis. Moreover, adipose OGT overexpression inhibits lipolysis and promotes diet-induced obesity. These findings establish an essential role for OGT in adipose tissue homeostasis and indicate a unique potential for targeting O-GlcNAc signaling in the treatment of obesity.

[1] Program in Integrative Cell Signaling and Neurobiology of Metabolism, Department of Comparative Medicine, Yale University School of Medicine, New Haven, CT 06510, USA. [2] Department of Cellular and Molecular Physiology, Yale University School of Medicine, New Haven, CT 06510, USA. *email: xiaoyong.yang@yale.edu

O ver the past few decades, the worldwide availability of energy-dense food has increased dramatically. At the same time, obesity rates have been rapidly increasing with no signs of abating and no effective strategy to combat it. Obesity is closely related to type 2 diabetes (T2D), cardiovascular diseases, liver diseases, cancer, and chronic diseases such as osteoarthritis, sleep apnea, and depression[1–3]. Obesity has become a significant contributor to increased morbidity and mortality[4,5]. Obesity occurs when excess nutrients are consumed, converted to fat, and stored in adipose tissue. Besides the overall obesity, the anatomical location of fat also matters in disease development. Studies have shown that intra-abdominal visceral fat is strongly positively correlated with obesity-related diseases, while peripheral subcutaneous fat seems to have no effect or even a protective effect against developing metabolic syndrome[6–10]. However, the biological and genetic differences between visceral and subcutaneous fat are poorly understood. A better understanding of the molecular and cellular mechanisms underlying the differential regulation of visceral and subcutaneous fat will enable us to develop novel approaches to effectively treat metabolically unhealthy obesity and potentially benefit the treatment of obesity-related diseases.

Once considered a passive lipid storage depot, adipose tissue is now known to have enormous plasticity and is capable of adapting its size, phenotype, and metabolic functions to nutrient availability[11–14]. These dynamics enable adipose tissue to store excess nutrients as fat and to provide energy by releasing fat through lipolysis. However, despite the increasingly recognized importance of adipose tissue homeostasis and its contribution to whole-body metabolism, how adipose tissue homeostasis is maintained, and how it responds to nutrient availability, are still largely unknown.

Nutrient flux into the hexosamine biosynthetic pathway leads to the post-translational modification of nuclear, cytoplasmic, and mitochondrial proteins by O-linked β-N-acetylglucosamine (O-GlcNAc) moieties. A single pair of enzyme control the cycling of O-GlcNAcylation: O-GlcNAc transferase (OGT) and O-GlcNAcase (OGA) catalyzing the addition and removal of O-GlcNAc moieties, respectively. The O-GlcNAc modification is proposed to be a nutrient-sensing mechanism since protein O-GlcNAcylation levels in cells fluctuate with the availability of nutrients[15–18]. Dysregulated O-GlcNAc signaling has been linked to perturbed fat storage in Caenorhabditis elegans and Drosophila[19–21]. A short isoform of OGA has been shown to accumulate on the surface of lipid droplets and modulates lipid droplet protein perilipin 2 (PLIN2) and PLIN3 levels in HeLa cells[22]. Despite the importance of O-GlcNAc signaling in lipid storage and lipid droplet protein regulation, the physiological function of O-GlcNAc signaling in adipose tissue homeostasis is poorly understood.

The present study focuses on the role of O-GlcNAc signaling in regulating adipose tissue dynamics in metabolic adaptation to nutrient availability. We show that visceral adipose tissue retains its fat mass by maintaining a high level of O-GlcNAcylation during acute fasting. Loss of adipose OGT specifically promotes lipolysis in visceral fat by decreasing O-GlcNAcylation and promoting phosphorylation of lipid droplet-associated PLIN1, while overexpression of adipose OGT inhibits adipose tissue lipolysis and promotes diet-induced obesity and whole-body insulin resistance. Our analyses also suggest that enhanced adipose O-GlcNAc signaling is a molecular signature for obesity and diabetes in humans. These data reveal an essential role for OGT in lipolysis regulation and indicate a unique potential for targeting O-GlcNAc signaling to combat metabolically unhealthy obesity.

## Results

**Adipose OGT deletion leads to a rapid loss of visceral fat.** To determine the role of OGT in adult adipose tissue homeostasis, we generated OGT adipose-specific knockout (AKO) mice by crossing a strain of mice harboring floxed alleles of Ogt with another strain in which the expression CreER, a tamoxifen-inducible Cre recombinase, is under the control of the adiponectin promoter (Fig. 1a). This model allowed for the depletion of OGT in various adipose depots, including interscapular brown adipose tissue (BAT), subcutaneous inguinal white adipose tissue (iWAT), and visceral epididymal white adipose tissue (eWAT) in adult mice (Fig. 1b). The knockout efficiency of Ogt in BAT, iWAT, and eWAT after tamoxifen administration was determined by quantitative real-time PCR (RT-PCR) analysis and Western blot analysis (Supplementary Fig. 1a, b and Fig. 1c). OGT AKO mice maintained a body weight similar to their wild-type (WT) littermate controls within 5 weeks after tamoxifen administration under normal chow (NC) feeding (Supplementary Fig. 1c). Metabolic cage analysis was performed 1 week after tamoxifen injection. The results showed that WT and OGT AKO mice had similar energy expenditure, rate of oxygen consumption ($VO_2$), food intake, and physical activity (Fig. 1d and Supplementary Fig. 2a–e). Interestingly, the respiratory exchange ratio (RER; $VCO_2/VO_2$) of OGT AKO mice was significantly lower than that of WT controls during the light cycle (Fig. 1e–g), suggesting that OGT ablation promotes whole-body utilization of lipids over carbohydrates. Quantitative magnetic resonance (QMR) analysis was performed 2 weeks after tamoxifen injection. A trend of decreasing fat mass and similar lean mass were observed in OGT AKO mice (Fig. 1h, i). To further explore the role of OGT in adipose tissue dynamics, mice were subjected to a 24 h fasting and 6 h refeeding treatment 2 weeks after tamoxifen administration. No difference in body weight was found between WT and OGT AKO mice during fasting/refeeding (Supplementary Fig. 1d). However, QMR analysis showed that fasting induced a more dramatic decrease in fat mass in OGT AKO mice, compared with WT controls (Fig. 1j). Further examination of various fat depots showed that compared to WT controls, OGT AKO mice had significantly decreased eWAT mass in fed status and fasting further exacerbated eWAT mass loss in OGT AKO mice. In contrast, no significant difference was found in tissue weights of BAT and iWAT between WT and OGT AKO mice (Fig. 1k–m). Together, these results reveal that OGT is essential for maintaining visceral fat mass, especially during fasting.

**Loss of adipose OGT promotes lipolysis in visceral fat.** To further determine the role of OGT in the fasting response of the adipose tissue, the dynamic change of fat depots during fasting/refeeding was examined. Strikingly, 6 h fasting induced a dramatic weight loss in eWAT in OGT AKO mice. In contrast, no significant change in eWAT weight was found in WT controls. Interestingly, 24 h fasting and 6 h refeeding induced similar eWAT weight changes in WT and OGT AKO mice (Fig. 2a, b). These data suggest that OGT ablation sensitizes visceral fat to acute fasting-induced eWAT weight loss, but does not affect its maximal response to prolonged fasting and its ability to restore its tissue weight during refeeding. No differences in BAT and iWAT weights between WT and OGT AKO mice were found during fasting/refeeding (Supplementary Fig. 3a, b). To determine the dynamic change of adipocytes, hematoxylin–eosin (H&E) staining was performed and the quantification of adipocyte size in eWAT showed that fasting induced a dramatic increase in the percentage of small-sized adipocytes in OGT AKO mice, while only a moderate effect was observed in WT

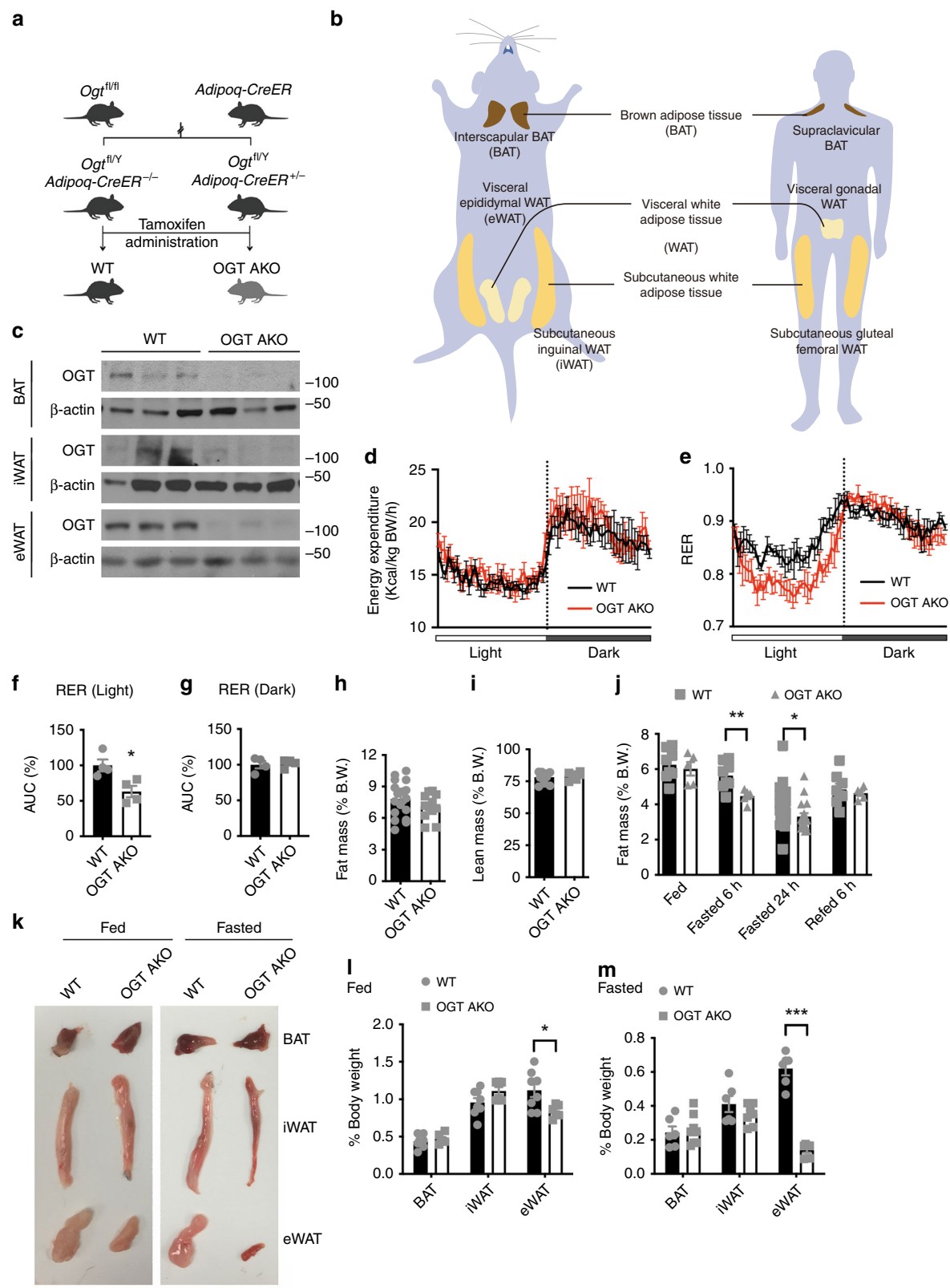

mice (Fig. 2c, d). Consistently, the average size of adipocytes in eWAT was significantly smaller in fasted OGT AKO mice, compared with WT controls (Fig. 2e). Meanwhile, no difference in iWAT adipocyte size was found between WT and OGT AKO mice (Supplementary Fig. 3c, d). These results reveal that loss of OGT specifically promotes fasting-induced adipocyte shrinkage

in eWAT. The analysis of blood samples showed that compared to WT controls, OGT AKO mice had a significantly higher serum free fatty acid (FFA) level after 6 h of fasting (Fig. 2f). In addition, ex vivo lipolysis assay showed that loss of OGT enhanced CL-316,243 (a β-3 adrenergic agonist)-induced gly-cerol release in eWAT, but not iWAT or brown adipocytes

**Fig. 1 Inducible deletion of adipose OGT leads to enhanced lipid utilization and a rapid loss of visceral fat. a** Breeding strategy used to generate wild-type (WT) control mice and adipose-specific *Ogt*-knockout (OGT AKO) mice. **b** Schematic diagram showing the localization of mouse adipose tissues including BAT, iWAT, and eWAT; the localization of corresponding human adipose tissues was shown as a reference. **c** Western blot analysis of OGT and β-actin in various adipose tissues from WT and OGT AKO mice. **d**, **e** Energy expenditure and respiratory exchange ratios (RERs) of WT and OGT AKO mice ($n = 4$ WT, 4 KO). **f**, **g** Area under curve (AUC) analyses of RER results shown in **e**. **h**, **i** Fat mass and lean mass of WT and OGT AKO mice fed on a normal chow (NC) ($n = 19$ WT, 11 KO for **h**; $n = 11$ WT, 6 KO for **i**). **j** Dynamic change of fat mass of WT and OGT AKO mice during fasting/refeeding ($n = 23$ WT, 18 KO for fasted 24 h; $n = 8$ WT, 5 KO for other groups). **k** Representative images of BAT, iWAT, and eWAT from WT and OGT AKO mice in fed and 24 h fasted states, scale bar is 1 cm. **l**, **m** Tissue weights, presented as % of body weight, in mice described in **k** ($n = 8$ WT, 6 KO for **l**; $n = 6$ WT, 8 KO for **m**). Data are presented as mean ± s.e.m. Statistical analysis: Student's *t* test, $*p < 0.05$, $**p < 0.01$, and $***p < 0.001$. Source data are provided as a Source Data file.

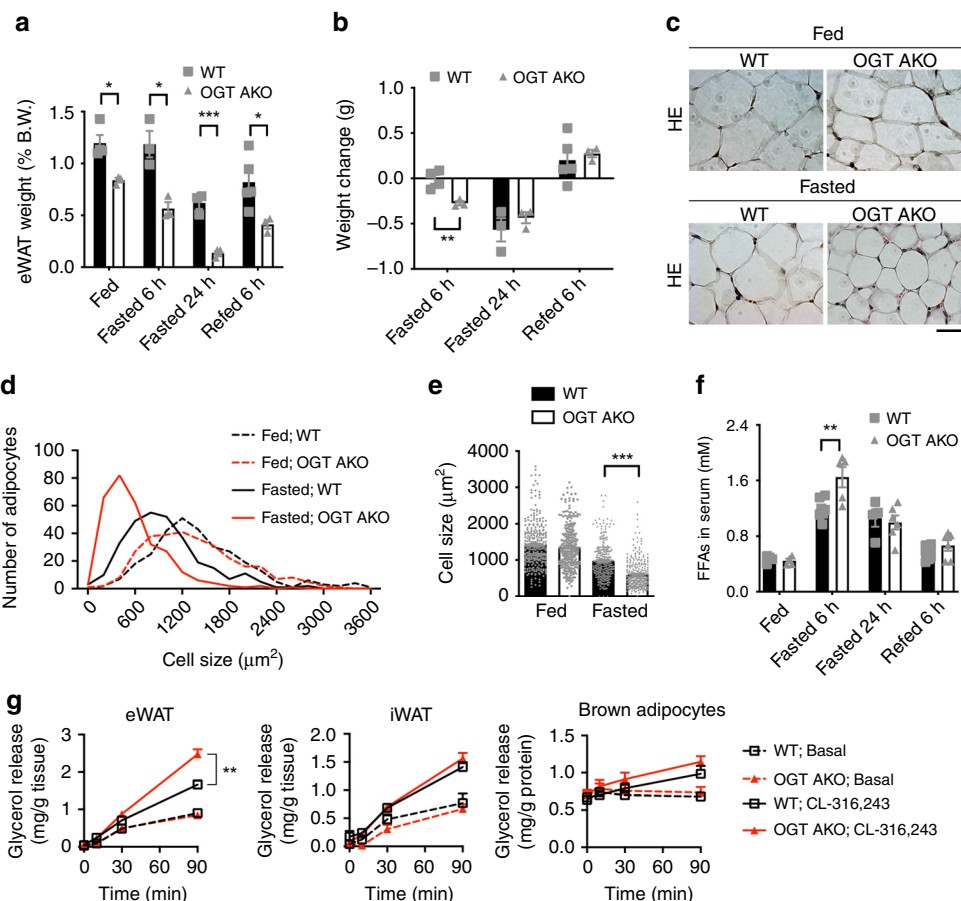

**Fig. 2 Loss of adipose OGT promotes lipolysis in visceral fat. a** Tissue weight of eWAT from WT and OGT AKO mice in fed, 6 h fasted, 24 h fasted, and 6 h refed states ($n = 3$–5 WT, 3–4 KO). **b** Changes in eWAT tissue weight during fasting/refeeding ($n = 3$–5 WT, 3 KO). **c** Hematoxylin–eosin (H&E) staining of eWAT from 2-month-old WT and OGT AKO mice in fed and 24 h fasted states, scale bar is 20 µm. **d**, **e** Quantification of adipocyte size in eWAT shown in **c** (three experiments). **f** Dynamic changes in serum FFA level in WT and OGT AKO mice during fasting/refeeding ($n = 4$–8 WT, 4–6 KO). **g** Basal and stimulated (10 µM CL-316,243) lipolysis measured by glycerol released from explants of eWAT and iWAT and isolated brown adipocytes from WT and OGT AKO mice ($n = 2$–6/group, three experiments). Data are presented as mean ± s.e.m. Statistical analysis: Student's *t* test, $*p < 0.05$, $**p < 0.01$, and $***p < 0.001$. Source data are provided as a Source Data file.

(Fig. 2g). Collectively, the above results demonstrate that loss of adipose OGT promotes fasting-induced lipolysis specifically in visceral fat.

Increased FFA flux into the circulation has been closely related to hepatic lipid accumulation and whole-body insulin resistance[23,24]. We measured the levels of triglycerides in the liver and observed a mild increase in hepatic triglyceride level in OGT AKO mice after 6 h of fasting. However, the fasting-induced hepatic triglyceride accumulation in OGT AKO mice was eliminated after 24 h of refeeding. Moreover, WT and OGT AKO mice had similar hepatic triglyceride levels after 8 weeks of

high-fat diet (HFD) feeding (Supplementary Fig. 4a). Also, glucose tolerance test (GTT) and insulin tolerance test (ITT) showed no difference between NC-fed WT and OGT AKO mice (Supplementary Fig. 4b–e). Together, these results suggest that the enhanced lipolysis in OGT AKO mice does not affect whole-body glucose metabolism and insulin sensitivity.

**O-GlcNAc signaling suppresses lipolysis in cultured cells.** We then tested the role of O-GlcNAc signaling in lipolysis in cultured cells. C3H/10T1/2 cells were differentiated into mature adipocytes and used for lipolysis study. Oleic acid (OA) treat-

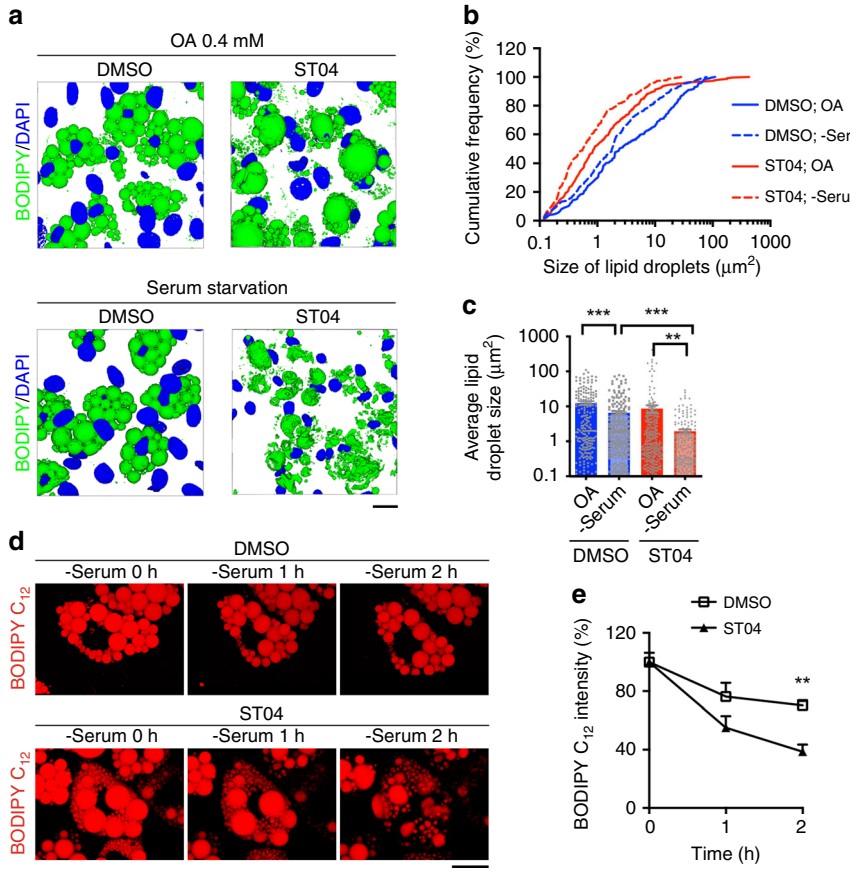

**Fig. 3 Inhibiting *O*-GlcNAc signaling enhances lipolysis in cultured cells. a** 3D-reconstructed confocal images of differentiated C3H/10T1/2 cells treated with DMSO or ST045849 (ST04), an OGT inhibitor, cultured in 0.4 mM oleic acid-supplemented medium and serum-depleted medium, and stained with BODIPY 493/503 (lipid droplets, green), and DAPI (nucleus, blue), scale bar is 20 μm. **b, c** Cumulative frequency distribution and mean average of lipid droplet size shown in **a** (three experiments). **d** Time-lapse imaging of differentiated C3H/10T1/2 cells treated with DMSO and ST04, and labeled with BODIPY 558/568 C12 fatty acid (lipid droplets, red), scale bar is 20 μm.; **e** Quantification of lipid droplet size shown in **d** ($n = 3$/group, three experiments). Data are presented as mean ± s.e.m. Statistical analysis: ANOVA with Dunnett multiple comparisons for **c** and Student's $t$ test for **e**, **$p < 0.01$, and ***$p < 0.001$. Source data are provided as a Source Data file.

ment and serum starvation were used to induce lipogenesis and lipolysis, respectively. Pharmacological inhibition of OGT using ST045849 (ST04)[25] significantly suppressed overall *O*-GlcNAcylation and induced an increase in the number of small-sized lipid droplets and an appearance of very few super-sized lipid droplets in OA-treated cells (Supplementary Fig. 5a and Fig. 3a–c). More importantly, serum starvation induced a dramatic shrinkage in lipid droplets in ST04-treated cells, while only a moderate lipid droplet shrinkage was observed in dimethyl sulfoxide-treated control cells (Fig. 3a–c). Time-lapse microscopy of differentiated C3H/10T1/2 cells also showed that inhibiting OGT enhanced serum starvation-induced lipid mobilization from lipid droplets (Fig. 3d, e). HeLa cells loaded with OA for 24 h were also used to study the role of *O*-GlcNAc signaling in lipolysis. The results showed that enhancing *O*-GlcNAc signaling in HeLa cells by overexpressing OGT ablated serum starvation-induced lipid droplet shrinkage while down-regulating *O*-GlcNAc signaling by overexpressing OGA promoted serum starvation-induced lipolysis in HeLa cells (Supplementary Fig. 5b–h). Finally, primary adipocytes differentiated in vitro from mouse eWAT stromal vascular fraction (SVF) were used. Bright field microscopy showed that enhancing *O*-GlcNAc signaling by thiamet-G (TMG, an OGA inhibitor) treatment suppressed forskolin (Fsk, a cAMP activator that stimulates lipolysis)-induced lipid droplet shrinkage in primary adipocytes (Supplementary Fig. 5i, j). Altogether, these

data suggest that *O*-GlcNAc signaling protects adipocytes from lipolysis in a cell-autonomous manner.

**Loss of OGT promotes PLIN1 phosphorylation in visceral fat.** To gain further mechanistic insight into the regulation of lipolysis by OGT, we first analyzed the expression levels of lipolysis regulators in various adipose tissues from fed and fasted WT and OGT AKO mice. RT-PCR analysis showed that the expression of genes involved in lipolysis (adipocyte triglyceride lipase, *Atgl*; hormone-sensitive lipase, *Hsl*; comparative gene identification-58, *CGI-58*; perilipin family proteins *Plin1–5*), fatty acid oxidation (very-long-chain acyl-CoA dehydrogenase, *Vlcad*; peroxisome proliferator-activated receptor gamma coactivator 1-α, *Pgc-1α*), and triglyceride synthesis (diacylglycerol acyltransferases *Dgat1* and *Dgat2*) in iWAT and eWAT were comparable between WT and OGT AKO mice, under both fed and fasted conditions (Supplementary Fig. 6a–l). We did observe significant increased expression of genes involved in de novo lipogenesis including acetyl-CoA carboxylase 1 (*Acc1*) and fatty acid synthase (*Fasn*) in OGT-knockout adipose tissues under fed and/or fasted conditions, but not during refeeding (Supplementary Fig. 6m–o). The messenger RNA (mRNA) levels of *Dgat1* and *Dgat2* also remained comparable between WT and OGT AKO mice during refeeding (Supplementary Fig. 6p). Moreover, comparable protein levels of ATGL, lipolysis regulator fat-specific protein 27 (Fsp27), PLIN family proteins (PLIN1–3), DGAT1, and DGAT2 were

detected in control and OGT small interfering RNA (siRNA)-treated HeLa cells (Supplementary Fig. 7a). The PLIN family proteins have been shown to coat lipid droplets and control lipolysis by regulating the access of cytoplasmic lipases to the lipid droplet surface[26–28]. To assess whether O-GlcNAc signaling affects the accumulation of PLIN family proteins on lipid droplets, we isolated lipid droplets from control and OGA-overexpressing HeLa cells. Western blot analysis showed comparable levels of PLIN1, PLIN2, and PLIN3 in lipid droplet fractions from control and OGA-overexpressing cells, suggesting that down-regulating O-GlcNAc signaling does not affect the lipid droplet accumulation of PLIN proteins (Supplementary Fig. 7b–e). Immunofluorescence staining also confirmed that the inhibition of O-GlcNAc signaling by ST04 did not affect the lipid droplet localization of PLIN1 (Supplementary Fig. 7f).

Previous studies have shown that protein kinase A (PKA)-mediated phosphorylation on HSL and PLIN1 contributes to the stimulation of lipolysis[29–32]. We then sought to determine the phosphorylation levels of HSL and PLIN1 in visceral fat during feeding and fasting. Western blot analysis showed that fasting induced similar increases in HSL phosphorylation level in WT and OGT AKO mice (Fig. 4a and Supplementary Fig. 8a–c). Enhanced Akt phosphorylation (serine 473) was observed in eWAT of fed, but not fasted, OGT AKO mice (Supplementary Fig. 8d, e). Fasting also induced an increase in PLIN1 protein level in both WT and OGT AKO mice (Fig. 4a). Strikingly, a clear mobility shift in PLIN1 was observed in the eWAT from fasted OGT AKO mice (Fig. 4a, arrowhead), suggesting that OGT may affect PLIN1 post-translational modification. Further examination by phos-tag gel analysis showed that loss of OGT in visceral fat increased the intensity of the upper band of PLIN1, which is considered as phosphorylated PLIN1. Moreover, treatment with lambda protein phosphatase (λ-PPase), which removes protein phosphate groups from serine, threonine, and tyrosine residues in phosphorylated proteins, significantly reduced the intensity of the upper band and eliminated the difference between WT and OGT AKO mice (Fig. 4b, c). Together, these results demonstrated that loss of OGT promotes PLIN1 phosphorylation in visceral fat. Next, we obtained two commercially available phospho-PLIN1 antibodies recognizing PLIN1 phosphorylation at serine 492 (p492) and serine 517 (p517) and examined PLIN1 phosphorylation in primary adipocytes differentiated from eWAT SVFs of WT and OGT AKO mice. Immunofluorescence staining showed that IBMX/Fsk (3-isobutyl-1-methylxanthine and Fsk, cAMP activators)-induced p492 and p517 PLIN1 phosphorylation were both significantly higher in OGT-knockout adipocytes, compared to WT adipocytes (Supplementary Fig. 8f and Fig. 4d–f). Western blot analysis also confirmed that PLIN1 p517 phosphorylation in eWAT was enhanced in fasted OGT AKO mice (Fig. 4g, h). These results demonstrate that loss of OGT promotes fasting-induced PLIN1 phosphorylation in eWAT.

Interestingly, knocking out OGT did not further promote fasting-induced PLIN1 p517 phosphorylation in iWAT (Fig. 4g, h). Further analysis showed that similar to eWAT, OGT knockout did not affect HSL phosphorylation and PLIN1 protein level in iWAT (Supplementary Fig. 8g–j). Immunofluorescence staining showed that IBMX/Fsk-induced p492 and p517 PLIN1 phosphorylation were both enhanced in cultured adipocytes differentiated from iWAT SVF of OGT AKO mice (Supplementary Fig. 9a–d), suggesting that OGT-mediated regulation of PLIN1 phosphorylation is conserved in in vitro cultured epididymal and inguinal adipocytes. However, ex vivo lipolysis assay showed that loss of OGT enhanced IBMX/Fsk-induced lipolysis in eWAT, but not in iWAT (Supplementary Fig. 9e), which is consistent with our previous result showing that loss of OGT enhanced CL-316,243-induced lipolysis specifically in eWAT (Fig. 2g). The

inconsistent results from mouse tissue and cultured cells suggest that the unique lipolysis response in different white fat depots is somehow lost in the in vitro cultured cell system. We then examined the in vivo fasting response in different fat pads, and Western blot analysis showed that a high level of O-GlcNAcylation was maintained in eWAT during fasting, while the overall O-GlcNAcylation level in iWAT was significantly lower and it rapidly dropped during fasting (Fig. 4i, j). These results suggest that OGT is required for maintaining a stable level of O-GlcNAcylation in eWAT during fasting, which protects eWAT mass by inhibiting PLIN1 phosphorylation and lipolysis. We also obtained the levels of OGT and OGA transcripts in human subcutaneous fat and visceral fat from the Genotype-Tissue Expression (GTEx) database[33,34]. Our analysis showed that the ratio of Ogt/Oga full-length transcripts was significantly higher in visceral fat than subcutaneous fat in both men and women (Fig. 4k). We also observed that women have a lower Ogt/Oga ratio than men for both subcutaneous and visceral fat (Fig. 4k). Moreover, an increase in the Ogt/Oga ratio during aging was found, especially in visceral fat from men (Supplementary Fig. 9f, g). These results suggest that the increased O-GlcNAcylation may also serve as a molecular signature for visceral fat in humans and it may contribute to the gender and age differences in adipose tissue homeostasis.

**OGT catalyzes PLIN1 O-GlcNAcylation.** Next, we sought to explore the molecular mechanism of OGT-mediated regulation of PLIN1 phosphorylation. Since the crosstalk between O-GlcNAcylation and phosphorylation has been reported in various studies[35,36], we tested the possibility that OGT may regulate PLIN1 phosphorylation by directly modifying PLIN1 by O-GlcNAcylation. Hemagglutinin (HA)-tagged PLIN1 was overexpressed with Myc-tagged OGT or a catalytic dead OGT mutant (OGT-CD) in HeLa cells. Then, immunoprecipitation (IP) analysis was performed and a significant level of O-GlcNAcylation on PLIN1 was observed in OGT-overexpressing cells. Moreover, the CD mutation of OGT severely impaired its activity to promote PLIN1 O-GlcNAcylation (Fig. 5a). These results demonstrated that OGT modifies PLIN1 by O-GlcNAcylation. To identify the glycosylation sites on PLIN1, we first generated a candidate list of potential glycosylation sites. The known PKA phosphorylation sites (S81, S222, S276, S433, S492, S517), which are phosphorylated during fasting, were included in the list[37,38]. We also ran three independent computational prediction analyses and obtained five potential glycosylation sites shown in all predictions. Among these sites, two of them (S433 and S492) are known PKA sites. Two other sites (S272 and S511) were also included since they are localized adjacent to the known PKA sites (Fig. 5b, c). Then, a series of PLIN1 mutations were generated and the effects of these mutations on overall PLIN1 O-GlcNAcylation were determined. The results showed that mutating serine 492 to alanine (S492A) and serine 517 to alanine (S517A) both significantly decreased the overall PLIN1 O-GlcNAcylation (Fig. 5d, e). We then generated the S492A/S517A double mutant PLIN1 (PLIN1-AA) and showed that the double mutation largely eliminated the overall O-GlcNAcylation on PLIN1 (Fig. 5f, g). The above data reveal that S492 and S517 are the primary PLIN1 O-GlcNAcylation sites, which are also known as PLIN1 phosphorylation sites closely involved in the regulation of lipolysis.

**OGT suppresses lipolysis by modulating PLIN1 function.** The above results suggest that PLIN1 can be competitively modified by O-GlcNAc and phosphate at S492 and S517. Supporting this idea, OGT overexpression promoted PLIN1 O-GlcNAcylation and decreased its overall serine phosphorylation in IBMX/Fsk-

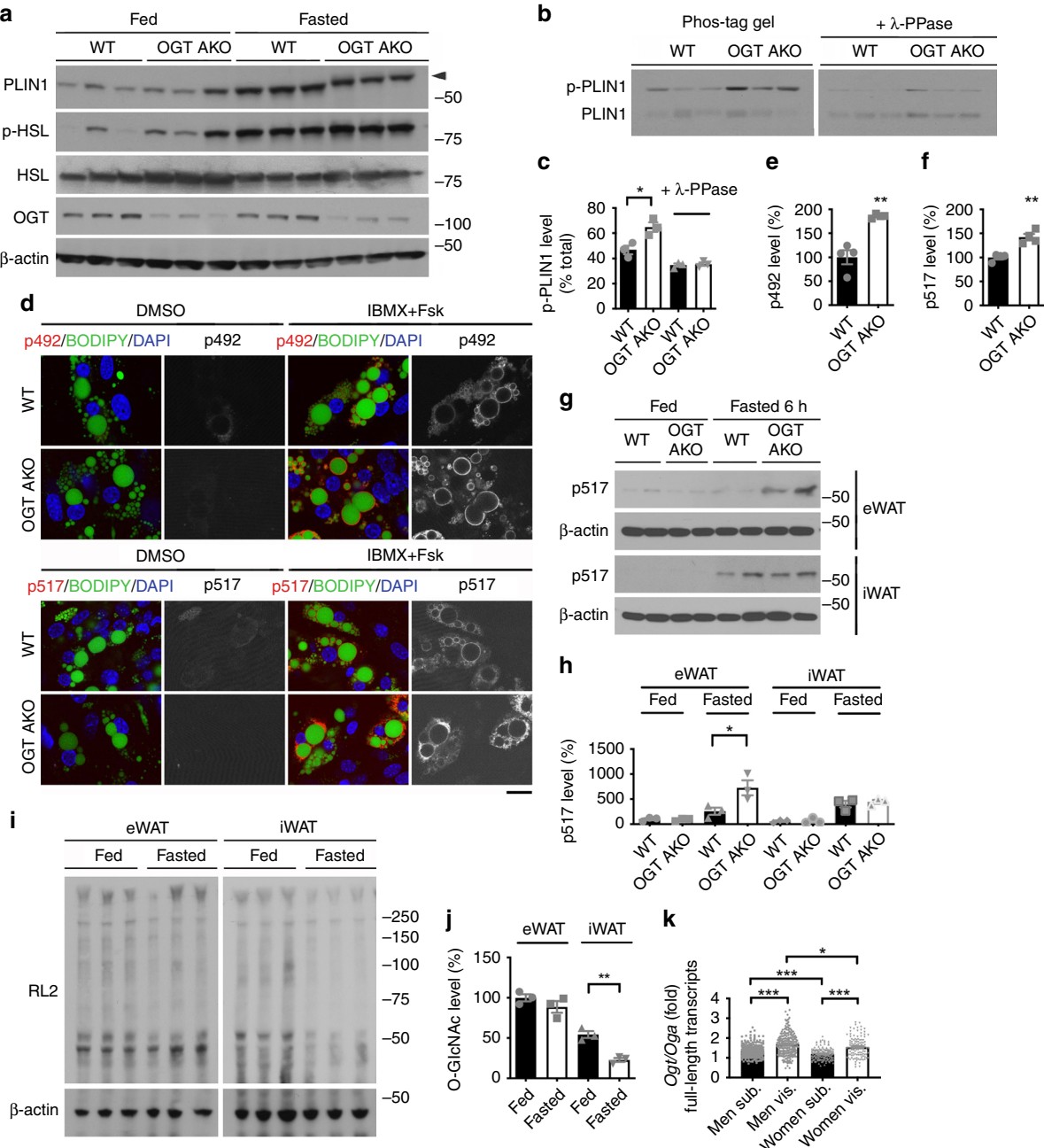

**Fig. 4 Loss of OGT promotes fasting-induced PLIN1 phosphorylation in visceral fat. a** Western blot analysis of PLIN1, p-HSL, HSL, OGT, and β-actin in eWAT of fed and 6 h fasted WT and OGT AKO mice (three experiments). **b**, **c** Phos-tag gel analysis and quantification of PLIN1 in eWAT of fed and 6 h fasted WT and OGT AKO mice; λ-PPase was used to remove phosphate groups from phosphorylated serine, threonine, and tyrosine residues ($n = 3$/group for **c**, three experiments). **d–f** Immunofluorescence staining and quantification of PLIN1 phosphorylation at serine 492 (p492) and serine 517 (p517) in primary adipocytes differentiated in vitro from the WT and OGT AKO eWAT SVFs ($n = 4$/group, three experiments); IBMX and forskolin (Fsk) were used to stimulate cAMP/PKA pathway; lipid droplets were stained with BODIPY 493/503 (green) and nuclei were stained with DAPI (blue), scale bar is 20 μm. **g**, **h** Representative Western blots and quantification of PLIN1 phosphorylation at serine 517 (p517) in eWAT and iWAT of fed and 6 h fasted WT and OGT AKO mice ($n = 3$/group for **h**, three experiments). **i**, **j** Western blot analysis and quantification of overall O-GlcNAcylation levels in eWAT and iWAT of fed and 6 h fasted WT and OGT AKO mice; RL2 recognizes O-GlcNAc modification on proteins ($n = 3$/group, three experiments). **k** Ratios of *Ogt/Oga* mRNA levels (full-length transcripts) in human subcutaneous fat (Sub.) and visceral fat (Vis.); original raw data was from the Genotype-Tissue Expression (GTEx) database ($n > 100$/group). Data are presented as mean ± s.e.m. Statistical analysis: ANOVA with Dunnett's multiple comparisons for **k** and Student's *t* test for the rest, \**p* < 0.05, \*\**p* < 0.01, and \*\*\**p* < 0.001. Source data are provided as a Source Data file.

treated HeLa cells (Fig. 6a). Similarly, decreased O-GlcNAcylation and increased phosphorylation (S517) of endogenous PLIN1 in eWAT was found in OGT AKO mice, compared to WT control mice (Fig. 6b). Further experiment showed that OGT overexpression abolished IBMX/Fsk-stimulated PLIN1

phosphorylation at S517 (Fig. 6c). Lipolysis assay also showed that OGT overexpression largely abrogated IBMX/Fsk-induced lipolysis in 293T cells (Fig. 6d). Next, we determined the role of PLIN1 S492/S517 phosphorylation in lipolysis by assessing the effects of non-phosphorylatable mutant PLIN1-AA and

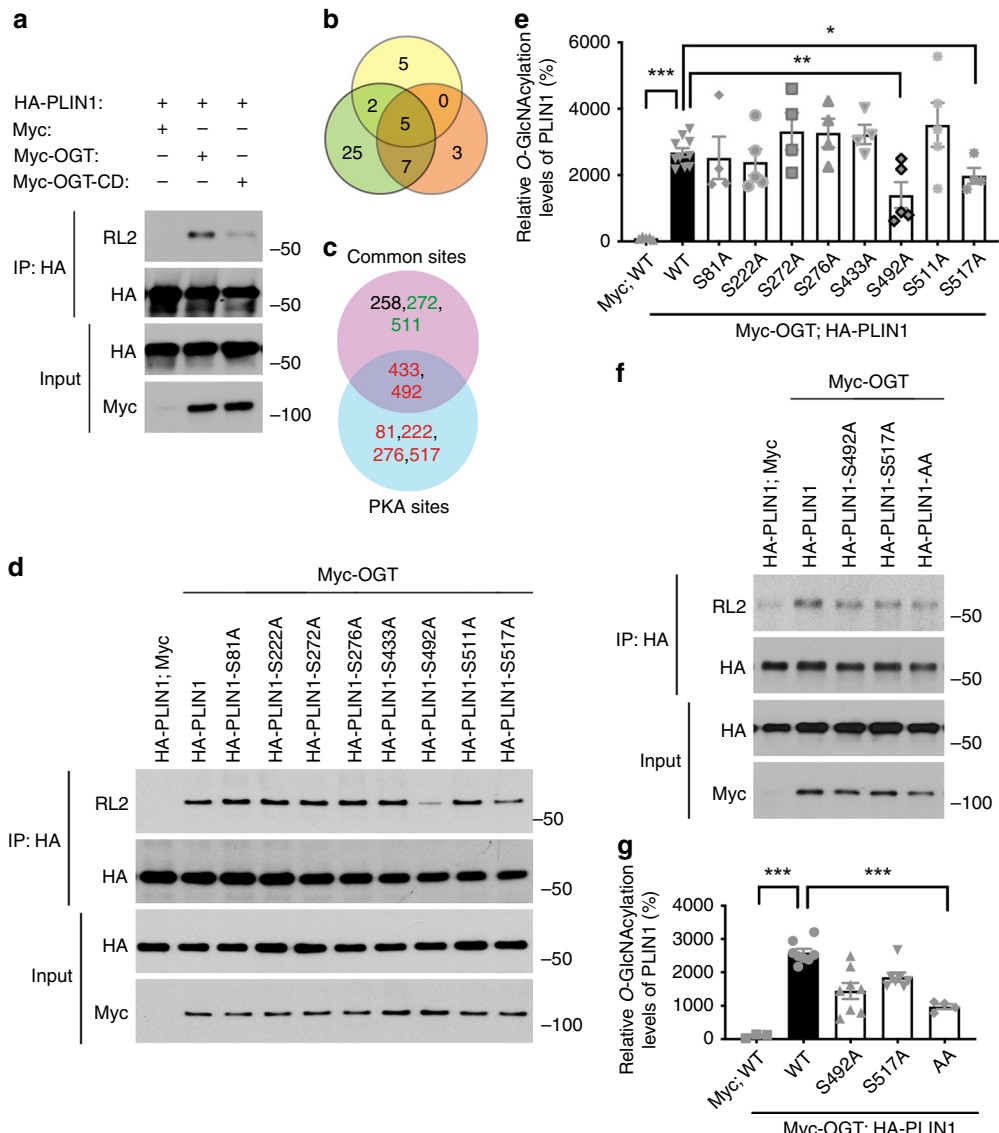

**Fig. 5 OGT catalyzes PLIN1 O-GlcNAcylation. a** Immunoprecipitation (IP) and Western blot analysis showing that OGT overexpression enhances PLIN1 *O*-GlcNAcylation and the catalytic dead mutation of OGT greatly impairs its ability to catalyze PLIN1 *O*-GlcNAcylation; RL2 is an antibody against *O*-GlcNAc moieties. **b** Overlapping of predicted *O*-GlcNAcylation sites determined by three independent analyses. **c** Final candidate sites, known PKA phosphorylation sites (red colored) were included, common sites from **b** that are adjacent to the PKA sites (green colored) were also included. **d**, **e** IP, Western blot analysis, and quantification results showing that OGT catalyzes PLIN1 *O*-GlcNAcylation primarily at serine 492 and serine 517 (*n* = 10 for WT positive control, 4–5 for other groups). **f**, **g** IP, Western blot analysis, and quantification results showing that serine 492 to alanine and serine 517 to alanine double mutation in PLIN1 (PLIN1-AA) largely eliminated its overall *O*-GlcNAcylation (*n* = 8 for WT positive control, 3–8 for other groups); data are presented as mean ± s.e.m. Statistical analysis: ANOVA with Dunnett's multiple comparisons, *\*p* < 0.05, *\*\*p* < 0.01, and *\*\*\*p* < 0.001. Source data are provided as a Source Data file.

phosphomimetic mutant PLIN1-S492E/S517E (PLIN1-EE) on IBMX/Fsk-induced lipolysis. The results showed that the non-phosphorylatable PLIN1-AA mutation decreased, while the phosphomimetic PLIN1-EE mutation increased lipolysis, demonstrating that PLIN1 S492/S517 phosphorylation promotes lipolysis in 293T cells (Fig. 6e). To further test whether PLIN1 phosphorylation is involved in OGT-mediated regulation of lipolysis, OGT was overexpressed together with PLIN1 or its mutants in 293T cells and their effects on basal and IBMX/Fsk-induced lipolysis were determined. Consistent with previous results, IBMX/Fsk induced a significant increase in lipolysis and OGT overexpression abolished IBMX/Fsk-stimulated lipolysis. PLIN1 overexpression had no significant effect on basal and stimulated lipolysis, while PLIN1-AA overexpression suppressed both basal and stimulated lipolysis. Moreover, PLIN1-EE

overexpression had no significant effect on basal lipolysis but largely rescued OGT-mediated repression of stimulated lipolysis (Fig. 6f). These results thus demonstrate that PLIN1 S492/S517 phosphorylation is required for both basal and stimulated lipolysis. It is worth noting that the phosphomimetic PLIN1-EE mutation itself was not sufficient to fully activate lipolysis (Fig. 6f), suggesting that PLIN1 phosphorylation is essential but not sufficient for the full activation of stimulated lipolysis. Nevertheless, these results demonstrate that the major effect of OGT-mediated suppression of lipolysis is through inhibiting PLIN1 S492/S517 phosphorylation. Immunofluorescence staining also showed that IBMX/Fsk treatment induced a significant lipid droplet shrinkage in PLIN1-overexpressing HeLa cells, which is shown by the increase in the percentage of small-sized lipid droplets and the decrease in the average size of lipid droplets.

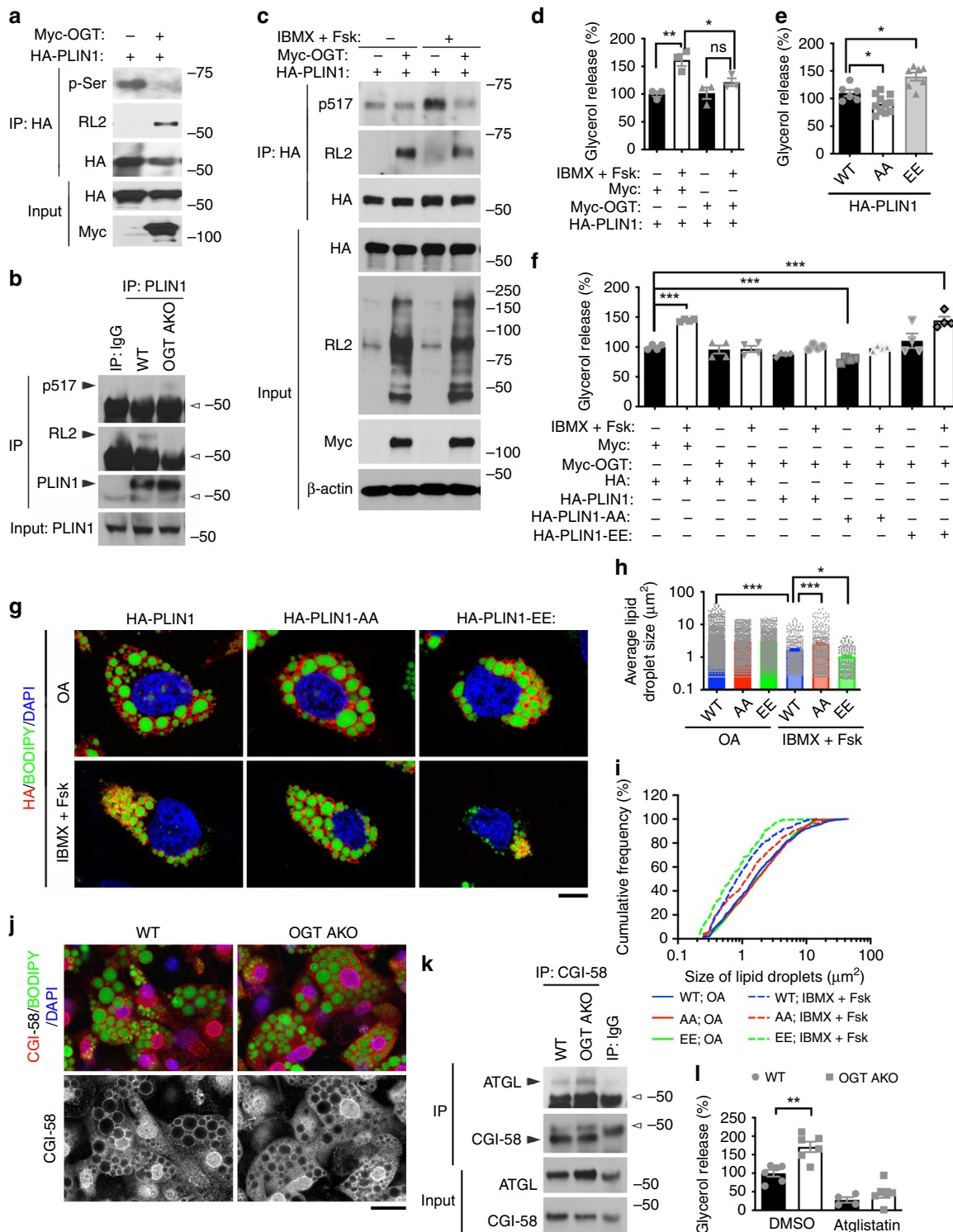

More importantly, compared to PLIN1-overexpressing cells, a much milder lipid droplet shrinkage was found in PLIN1-AA-overexpressing cells, while a more dramatic shrinkage was observed in PLIN1-EE-overexpressing cells (Fig. 6g–i). Again, these results demonstrate that OGT-mediated inhibition of PLIN1 phosphorylation is essential for its role in suppressing IBMX/Fsk-stimulated lipolysis.

Previous studies have shown that during stimulated lipolysis, phosphorylated PLIN1 will disassociate with CGI-58, which allows CGI-58 to bind and activate ATGL, the rate-limiting enzyme in lipolysis[39,40]. Consistent with these findings, our

results showed that OGT knockout promoted the release of CGI-58 from lipid droplets and enhanced the interaction between CGI-58 and ATGL upon lipolytic stimulation (Fig. 6j, k). Moreover, inhibition of ATGL activity using atglistatin abolished the difference in IBMX/Fsk-stimulated lipolysis between WT and OGT-knockout adipocytes (Fig. 6l), demonstrating that the effect of OGT on lipolysis is dependent on ATGL-mediated lipolysis.

**Adipose OGT suppresses lipolysis and promotes obesity.** To further explore the role of OGT in adipose tissue homeostasis, we

**Fig. 6 OGT supresses lipolysis by modulating PLIN1 O-GlcNAcylation and phosphorylation. a** IP and Western blot analysis showing the effects of OGT on PLIN1 O-GlcNAcylation and phosphorylation; p-Ser antibody recognizes proteins phosphorylated on serine residues; cells were treated with IBMX/Fsk before analysis. **b** IP and Western blot analysis showing the effects of OGT knockout on endogenous PLIN1 O-GlcNAcylation and phosphorylation; hollow arrowheads indicate IgG heavy chains. **c** IP and Western blot analysis showing that OGT overexpression attenuates IBMX/Fsk-induced PLIN1 phosphorylation at serine 517. **d** IBMX/Fsk-induced lipolysis measured by glycerol release ($n = 3$/group). **e** IBMX/Fsk-induced lipolysis measured by glycerol released from cells transfected with HA-PLIN1 (WT), HA-PLIN1-AA (AA), and HA-PLIN1-EE (EE) ($n = 6$–11/group). **f** Basal and IBMX/Fsk-induced lipolysis measured by glycerol released from cells transfected with Myc or Myc-OGT together with HA, HA-PLIN1, HA-PLIN1-AA, or HA-PLIN1-EE ($n = 4$/group). **g** Immunofluorescence images of cells transfected with HA-PLIN1, HA-PLIN1-AA, or HA-PLIN1-EE, cultured in 0.4 mM oleic acid, treated with or without IBMX/Fsk, and stained with antibody against HA (Red), BODIPY 493/503 (green), and DAPI (blue); scale bar is 10 μm. **h**, **i** Mean average and cumulative frequency distribution of lipid droplet size shown in **g** ($n > 150$/group). **j** Immunofluorescence images of primary adipocytes differentiated from WT and OGT AKO eWAT SVFs; adipocytes were treated with IBMX/Fsk, fixed, and stained with antibody against CGI-58 (red), BODIPY 493/503 (green), and DAPI (blue), scale bar is 20 μm. **k** IP and Western blot analysis showing the interaction between endogenous CGI-58 and ATGL in eWAT of 6 h fasted WT and OGT AKO mice; hollow arrowheads indicate IgG heavy chains. **l** IBMX/Fsk-induced lipolysis measured by glycerol released from primary adipocytes differentiated from WT and OGT AKO eWAT SVFs ($n = 4$ for Atglistatin-treated WT, six for other groups); Atglistatin (10 μM) was used to inhibit ATGL activity. Data are presented as mean ± s.e.m. Statistical analysis: Student's $t$ test for **i** and ANOVA with Dunnett multiple comparisons for the rest, *$p < 0.05$, **$p < 0.01$, and ***$p < 0.001$; n.s., not significant. Source data are provided as a Source Data file.

generated OGT AKI mice, in which exogenous rat OGT (rOGT, shares 99% protein identity with mouse OGT) is inducibly expressed in adipose tissue of adult mice (Fig. 7a, b). OGT AKI mice showed similar body weight and body composition at 5 weeks after tamoxifen-induced rOGT expression (Supplementary Fig. 10a–c). Surprisingly, no difference in fasting-induced FFA flux into the circulation or adipocyte shrinkage was found in OGT AKI mice, compared to WT controls (Supplementary Fig. 10d–g). Ex vivo lipolysis assay also showed no differences in basal and CL-316,243-induced lipolysis in eWAT, iWAT, or brown adipocytes between WT and OGT AKI mice (Supplementary Fig. 9h). We then fed the mice with an HFD and found that OGT AKI mice gained more body weight during HFD feeding (Fig. 7c). QMR analysis showed that HFD-fed OGT AKI mice had increased fat mass but similar lean mass, compared to WT controls (Fig. 7d–f). Further examination showed that compared to WT controls, HFD-fed OGT AKI mice had significantly increased tissue weights in iWAT, eWAT, and retroperitoneal visceral WAT (Fig. 7g, h). These results demonstrate that OGT overexpression in adipose tissue promotes HFD-induced obesity in mice. IP assay showed that OGT knockin enhanced endogenous PLIN1 O-GlcNAcylation and suppressed its phosphorylation (S517) (Fig. 7i). Ex vivo lipolysis assay was performed again using WAT of HFD-fed WT and OGT AKI mice. The results showed that OGT overexpression largely abolished CL-316,243-induced lipolysis in eWAT and iWAT (Fig. 7j). Western blot analysis showed that HFD feeding dramatically enhanced overall O-GlcNAcylation level in white fat, especially in eWAT (Supplementary Fig. 11a). More importantly, under HFD feeding, OGT AKI further enhanced the overall O-GlcNAcylation level in both iWAT and eWAT (Supplementary Fig. 11b). These results suggest that OGT promotes PLIN1 O-GlcNAcylation in WAT during HFD feeding, which leads to the inhibition of lipolysis. Transcriptional levels of lipogenic genes including *Acc1*, *Fasn*, *Dgat1*, and *Dgat2* in visceral fat were similar between HFD-fed WT and OGT AKI mice (Supplementary Fig. 11c, d), suggesting that lipogenesis may not be affected by OGT knockin. ITT and GTT showed that HFD-fed OGT AKI mice were less glucose-tolerant and less insulin-sensitive than WT control mice (Fig. 7k–n), demonstrating that OGT overexpression in adipose tissue promotes HFD-induced insulin resistance in mice.

Altogether, our study demonstrates an essential role for OGT in adipose tissue lipolysis. Enhanced O-GlcNAc signaling promotes PLIN1 O-GlcNAcylation and inhibits lipolysis, whereas suppressed O-GlcNAc signaling favors PLIN1 phosphorylation and lipolysis (Fig. 7o). We then corroborated these findings in ~40 different mouse strains by correlating the activation of O-

GlcNAc signaling with metabolic phenotypes. The analyses of the datasets obtained from the GeneNetwork database showed that despite the marked genetic heterogeneity of these mouse strains, the levels of fasting serum FFA in HFD-fed mice were negatively correlated with the *Ogt/Oga* transcript ratio in white fat. A clear trend of negative correlation was also observed in NC-fed mice (Fig. 8a). Moreover, the trends of positive correlation between adipose *Ogt/Oga* ratio and glycemia in GTT (glucose area under curve, glucose AUC) were observed in both NC and HFD-fed mice ($p = 0.12$ for HFD-fed group) (Fig. 8b). These results suggest a role for adipose O-GlcNAc signaling in the regulation of lipolysis and whole-body metabolism across various mouse strains.

To gain further insight into the contribution of OGT-mediated regulation of lipolysis to human health, we obtained human phenotypic data, including body mass index (BMI) and diagnosis of diabetes from the dbGAP database. Our analysis showed that the *Ogt/Oga* transcript ratio in subcutaneous fat was positively related with BMI (Fig. 8c). Again, visceral fat generally had a higher *Ogt/Oga* ratio, but no significant correlation between the *Ogt/Oga* ratio and BMI was observed (Fig. 8d). Human subjects diagnosed with T2D had higher *Ogt/Oga* ratios than non-diabetic normal controls in both subcutaneous fat ($p < 0.0001$) and visceral fat ($p = 0.09$) (Fig. 8e, f). These results suggest that an increased adipose *Ogt/Oga* ratio is a molecular signature of obesity and diabetes in humans. Altogether, our results reveal that OGT is a risk factor for obesity and the elevation of O-GlcNAcylation in adipose tissue may serve as a critical link between nutrient surplus and whole-body metabolic dysfunction.

## Discussion
In our study, we employed the inducible deletion system, which allows us to determine the unique role of OGT in adipose tissue metabolism. We demonstrated that the primary role of OGT in adipose tissue is inhibiting stimulated lipolysis in visceral fat. Loss of adipose OGT enables a rapid loss of visceral fat by sensitizing visceral fat to lipolytic stimuli, while OGT overexpression in adipose tissue inhibits stimulated lipolysis and promotes HFD-induced white fat accumulation. We further showed that O-GlcNAc modification in visceral fat is strongly enhanced in response to excess nutrients and maintains a high level during fasting. Our results thus suggest that suppressing O-GlcNAc signaling over a short period may serve as a promising strategy to selectively target central obesity. Previously, an independent group observed that constitutive deletion of OGT in mouse adipose tissue leads to decreased white fat mass and lipid accumulation in brown fat[41]. Constitutive deletion of OGT specifically in

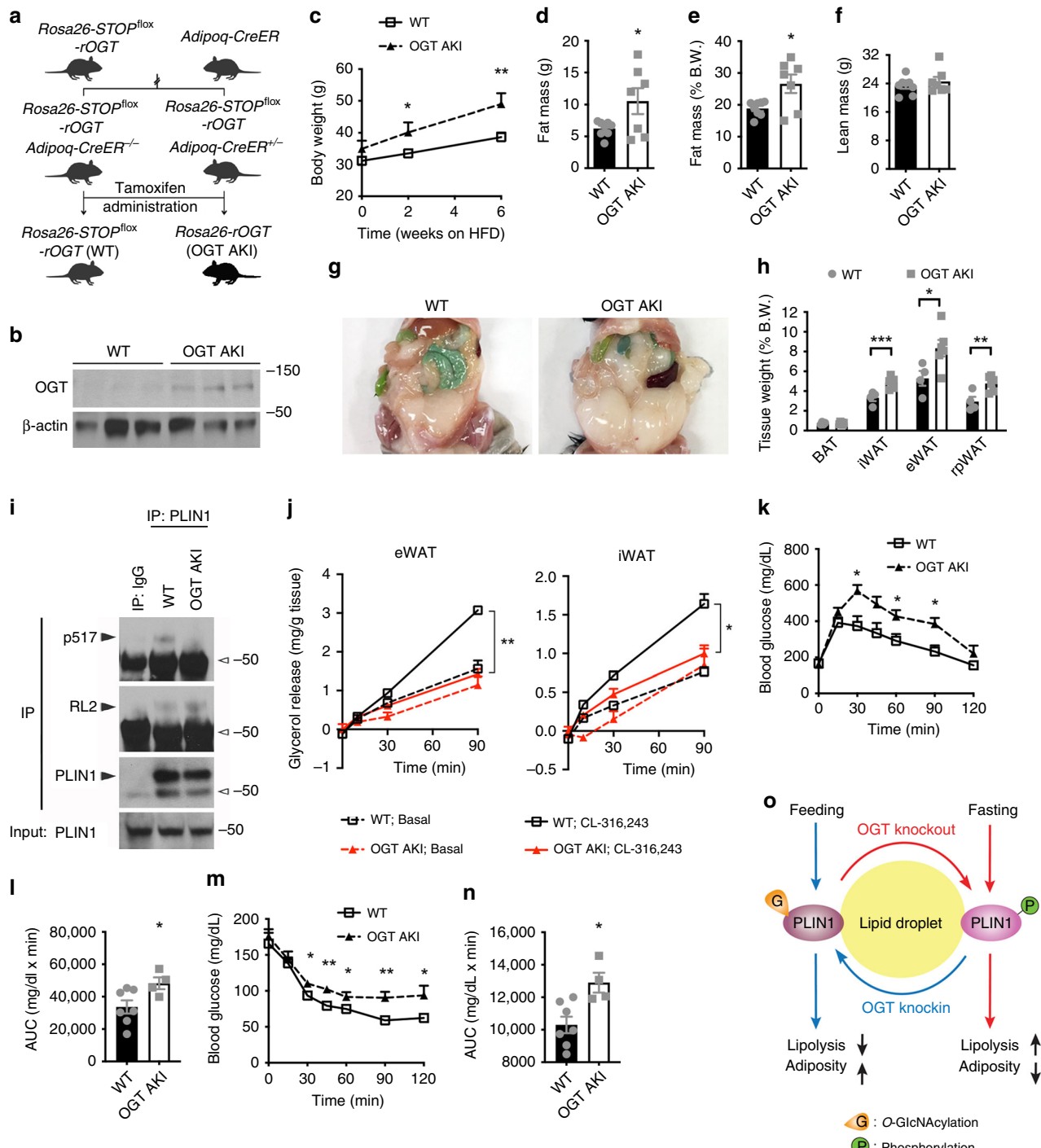

**Fig. 7 Adipose OGT suppresses lipolysis and promotes diet-induced obesity and insulin resistance. a** Breeding strategy used to generate WT control mice and OGT AKI mice. **b** Western blot analysis of OGT and β-actin in eWAT from WT and OGT AKI mice. **c** Body weight of WT and OGT AKI mice fed on HFD ($n = 11$ WT, 10 KI). **d–f** Fat mass and lean mass of WT and OGT AKI mice fed on HFD for 4 weeks ($n = 8$ WT, 7 KI). **g, h** Images and tissue weights of fat pads from HFD-fed WT and OGT AKI mice ($n = 4$ WT, 6 KO for **h**). **i** IP and Western blot analysis showing the effects of OGT knockin on endogenous PLIN1 O-GlcNAcylation and phosphorylation; hollow arrowheads indicate IgG heavy chains. **j** Basal and stimulated (10 μM CL-316,243) lipolysis measured by glycerol released from explants of eWAT and iWAT from HFD-fed WT and OGT AKI mice ($n = 2$/group, three experiments). **k–n** Blood glucose and area under curve (AUC) analyses of glucose tolerance test and insulin tolerance test of HFD-fed WT and OGT AKI mice ($n = 7$ WT, 4 KI). **o** Molecular model for OGT function in lipolysis. Data are presented as mean ± s.e.m. Statistical analysis: Student's $t$ test, *$p < 0.05$, **$p < 0.01$, and ***$p < 0.001$. Source data are provided as a Source Data file.

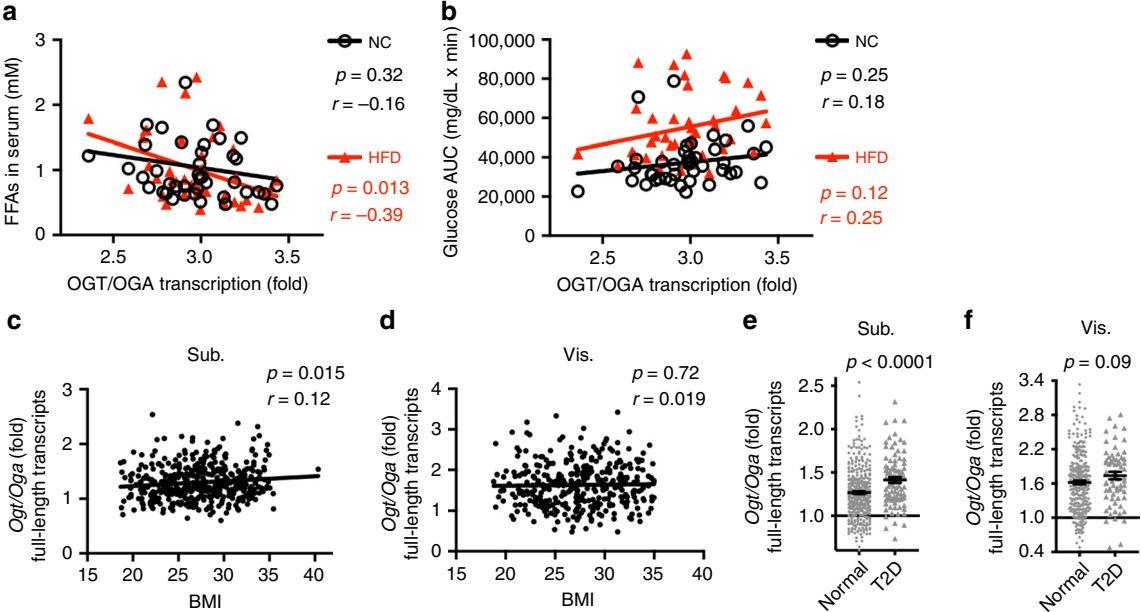

**Fig. 8 Increased adipose *Ogt/Oga* ratio is a molecular signature of impaired whole-body metabolism in mice and obesity and diabetes in humans. a** Correlation between the ratio of white fat *Ogt/Oga* transcript levels and serum free fatty acid (FFA) levels in 41 different mouse strains from the GeneNetwork database (the EPFL LISP3 Cohort); NC-fed and HFD-fed male mice at the fasted state were used for the analysis. **b** Correlation between the ratio of white fat *Ogt/Oga* transcript levels and glycemia during oral glucose tolerance test (glucose AUC); mice described in **a** were used. **c**, **d** Correlations between ratios of *Ogt/Oga* transcript levels (full-length transcripts) in human subcutaneous fat (Sub.) and visceral fat (Vis.) and body mass index (BMI); data from men and women were used for the analysis; original raw data was from the GTEx database (dbGaP study accession: phs000424.v7.p2) (n = 441 for **c**; n = 356 for **d**). **e**, **f** The average ratio of *Ogt/Oga* transcripts in subcutaneous fat (Sub.) and visceral fat (Vis.) of human subjects diagnosed with type 2 diabetes (T2D) and non-diabetic normal controls (n = 348 normal, 91 T2D for **e**; n = 286 normal, 68 T2D for **f**); data were obtained from the same study described in **c**, **d**. Data are presented as mean ± s.e.m. Statistical analysis: Student's *t* test for **e**, **f** and linear regression for the other panels. Source data are provided as a Source Data file.

BAT leads to decreased uncoupling protein 1 expression, impaired mitochondrial biogenesis, and cold intolerance[41]. These results suggest that adipose OGT has an essential role in the regulation of lipid metabolism.

An appealing idea in the field is that visceral fat accumulation is associated with elevated FFA flux into the circulation, which further leads to insulin resistance in various tissues[23,24]. Here we show that suppressing *O*-GlcNAc signaling in adipose tissue leads to a rapid loss of visceral fat by promoting lipolysis. However, no difference in whole-body insulin sensitivity was found in the OGT AKO mice. Similar results were also observed in an independent study[42]. These results suggest that the transient elevation in plasma FFA level in OGT AKO mice is not sufficient to cause whole-body insulin resistance. It is worth noting that a mild increase in hepatic triglyceride level in OGT AKO mice was observed only during fasting, which is consistent with the finding that loss of OGT only promotes stimulated lipolysis, but not basal lipolysis. Together, these findings suggest that inhibiting adipose *O*-GlcNAc signaling is an effective strategy to combat central obesity without noticeable adverse effects on whole-body metabolism. On the other hand, OGT overexpression in adipose tissue inhibits stimulated lipolysis and promotes HFD-induced white fat accumulation and insulin resistance. Our previous result showed that constitutive adipose-specific *Ogt*-knockout mice are resistant to diet-induced obesity[43]. These findings demonstrate that adipose OGT is essential for the dynamic adaptation of adipose tissue to nutritional cues. It also suggests that enlarged white fat can mediate whole-body insulin resistance through a lipolysis-independent mechanism.

In our study, the candidate gene approach was used to identify *O*-GlcNAcylation sites on PLIN1. Considering that our focus is to determine how PLIN1 *O*-GlcNAcylation affects lipolysis, we

included the PKA sites in the candidate list since they are phosphorylated in response to lipolytic stimuli. Computational prediction programs, YinOYang 1.2, OGlcNAcScan, and OGT-Site, were also used to provide more potential *O*-GlcNAcylation sites[44–46]. Studies have shown that both YinOYang 1.2 and OGlcNAcScan efficiently identified *O*-GlcNAc sites localized to the consensus sequence P-P-V-[ST]-T-A. But for sites located in a sequence differing from this consensus sequence, both programs generated numerous false-negative predictions[47]. OGTSite is a two-layered machine learning-based predictive model. Studies have demonstrated that it outperforms other *O*-GlcNAcylation site prediction programs by providing a ~84% accuracy[46]. By combining different approaches, we should be able to identify the essential *O*-GlcNAcylation sites related to the regulation of lipolysis. This is supported by the results showing that the S492A/S517A mutation decreased overall PLIN1 *O*-GlcNAcylation by ~60% and almost abolished IBMX/Fsk-induced lipolysis stimulation. However, it is possible that other *O*-GlcNAcylation sites on PLIN1 may also exist, especially in other tissues or under other physiological conditions. Other non-biased approaches, such as mass spectrometric analysis, should provide a more comprehensive understanding of PLIN1 *O*-GlcNAcylation. Nonetheless, our results strongly support a model in which OGT inhibits stimulated lipolysis by catalyzing PLIN1 *O*-GlcNAcylation at S492 and S517 and suppressing PLIN1 phosphorylation at the same sites.

It seems that the OGT-PLIN1-lipolysis axis is conserved among various cell types. The specific regulation of visceral fat lipolysis in OGT AKO mice can be at least partially attributed to the unique *O*-GlcNAc signaling dynamics in visceral fat. Our results show that visceral fat maintains a high *O*-GlcNAcylation level during acute fasting, and HFD feeding promotes *O*-

GlcNAcylation level in both fat depots with a stronger effect in visceral fat. Moreover, OGT overexpression further enhances white fat *O*-GlcNAcylation level in HFD-fed mice. These changes in *O*-GlcNAcylation level correlate with weight changes of white fat during fasting and HFD feeding. Thus, we conclude that *O*-GlcNAc modification is essential for maintaining visceral fat during fasting and promoting subcutaneous and visceral fat expansion during HFD feeding. Therefore, suppressing *O*-GlcNAc signaling might be a promising strategy to combat metabolically unhealthy obesity by targeting different fat pads in varying degrees.

## Methods

**Mice**. *Ogt*-floxed mice on the C57BL/6 background were kindly provided by Dr. Steven Jones at the University of Louisville[48]. Adipoq-CreER mice were from The Jackson Laboratory (Stock No: 024671). To generate the Rosa26-STOP^flox^-rOGT mice, a Flag-HA-rOGT-eGFP coding sequence was inserted into the STOP-eGFP-ROSA26TV plasmid (Addgene #11739, a.pngt from Klaus Rajewsky). The plasmid was linearized and electroporated into B6/BLU ES cells. Targeted ES cell clones were then used for blastocyst injection and the production of the Rosa26-STOP^flox^-rOGT mice. *Ogt*-floxed mice and Rosa26-STOP^flox^-rOGT mice were bred to Adipoq-CreER mice in the Yale Animal Resource Center to obtain OGT AKO and OGT AKI mice. To induce Cre recombinase activity, mice were injected intraperitoneally (i.p.) with 100 μL tamoxifen (Sigma-Aldrich, 20 mg/ml in vegetable oil) daily for 5 days. All mice were kept on a 12 h:12 h light–dark cycle. Mice were free to access water and fed with a NC diet or a 60% HFD (Research Diets, D12492) unless otherwise stated. All relevant ethical regulations for animal testing and research have been complied with. All animal studies received ethical approval from Yale University's Institutional Animal Care and Use Committee.

**Metabolic assays**. Body composition was examined using an EchoMRI system. For serum FFA level determination, tail-vein blood was collected at designated times and FFA levels were measured using an FFA Quantitation Kit (Sigma-Aldrich). For GTT, mice were fasted overnight and injected intraperitoneally with glucose (1.5 g/kg body weight), then tail-vein blood was collected at designated times, and blood glucose levels were measured using a Nova Max Glucometer. For ITT, mice were fasted for 6 h in the morning and injected intraperitoneally with insulin (1 U/kg body weight for NC-fed mice; 1.5 U/kg body weight for HFD-fed mice), then blood glucose levels were measured as described above. For metabolic cage analysis, mice were acclimated in metabolic chambers for 3 days and then gas exchange, energy expenditure, food intake, and physical activity were recorded continuously for 4 days.

**RNA and real-rime PCR**. Total RNA from mouse tissues was extracted using TRIzol reagent (Invitrogen). Complementary DNA (cDNA) samples were prepared using the iScript cDNA Synthesis Kit (Bio-Rad). Gene transcript levels were determined by amplifying the cDNA with SYBR Green Supermix (Bio-Rad) by using a LightCycler 480 real-time PCR machine (Roche). The expression of 36b4 was used as an internal control. Primer sequences are listed in Supplementary Table 1.

**Histology**. Mouse tissues were fixed in 4% paraformaldehyde overnight at 4 °C, dehydrated, and embedded in paraffin. Tissue sections were prepared by using a microtome and stained with H&E.

**Antibodies and chemicals**. Antibodies against OGT (ab96718), OGA (MGEA5, ab124807), *O*-GlcNAc (RL2, ab2739), PLIN1 (ab3526), SNAP23 (ab3340), ATGL (ab99532), DGAT1 (11561-1-AP), Fsp27 (CIDE C, ab77115), and p-Ser (phosphoserine, PSR-45) (Abcam); p492-phosphorylated PLIN1 (4855) and p517-phosphorylated PLIN1 (4856) (Vala Sciences); HA (H3663) (Sigma-Aldrich); CGI-58 (ABHD5, 12201-1-AP), PLIN2 (15294-1-AP), and PLIN3 (TIP47, 10694-1-AP) (Proteintech); p563-HSL (4139), phospho-Akt (Ser473, 9271), and Akt (9272) (Cell Signaling Technology); Myc (sc-40), DGAT2 (sc-66859), and β-actin (sc-8432) (Santa Cruz Biotechnology) were purchased from the indicated sources. Horse-radish peroxidase-conjugated secondary antibodies were from Santa Cruz Biotechnology. Alexa Fluor 594-conjugated secondary antibodies, BODIPY 493/503, and BODIPY 558/568 C12 fatty acid were obtained from Thermo Fisher Scientific. 4′,6-Diamidino-2-phenylindole (DAPI), OA, Fsk, and IBMX were from Sigma-Aldrich. CL-316,243 was from R&D Systems. OGT inhibitor ST045849 (ST04) was purchased from TimTec. TMG was from Cayman Chemical.

**Plasmids and siRNAs**. Mammalian expression plasmids for Myc-OGT and CD Myc-OGT-G598S (Myc-OGT-CD) were provided by Dr. Xiaochun Yu at the University of Michigan[49]. pCMV5-HA-PLIN1 was provided by Dr. Peng Li at Tsinghua University, China[50]. PLIN1 mutants were generated by site-directed mutagenesis using QuikChange II Site-directed Mutagenesis Kit (Agilent).

Scrambled siRNA (5′-GAGGCAUGUCCGUUGAUUCGU-3′) and OGT siRNA (5′-GAGGCAGUUCGCUUGUAUCGU-3′) were synthesized by Dharmacon.

**Cell culture and transfection**. C3H/10T1/2, HeLa, and 293T cells were from the American Type Culture Collection (ATCC) and were grown in Dulbecco's modified Eagle's medium supplemented with 10% fetal bovine serum at 37 °C in 5% CO₂. OGA-Tet off HeLa stable cell line was established in our laboratory. C3H/10T1/2, HeLa, and 293T cells were tested mycoplasma-free. C3H/10T1/2 cells were differentiated into adipocytes as described but without BMP-7 treatment[51]. For primary adipocytes, SVF isolated from inguinal or epididymal white fat were cultured and differentiated into mature adipocytes[52,53]. Briefly, freshly dissected white fat was cut into ~2 mm diameter pieces, digested at 37 °C for 40 min in calcium- and magnesium-free PBS supplemented with 10 mg/mL collagenase D (Roche, 11088882001), 2.4 mg/mL dispase II (Roche, 04942078001), and 1 mM calcium chloride. Digested tissue was then filtered through a 70 μm membrane and centrifuged at 500 × g for 10 min. The pelleted SVF fraction was then resuspended and plated onto collagen-coated plates. When cells reached 100% confluence, a cocktail containing 0.5 mM IBMX, 1 μM dexamethasone, 850 nM insulin, and 1 μM rosiglitazone was used to induce differentiation. After 48 h, media were switched to contain 850 nM insulin and 1 μM rosiglitazone. Media were changed on day 4 and only 850 nM insulin was continued. Same media were given again on day 6. Differentiated adipocytes on day 7 were used for further experiments. Treatment with 0.4 mM OA was used to promote lipid droplet formation and serum starvation or IBMX/Fsk treatment was used to induce lipolysis. All plasmids were transfected into cells using FuGENE HD (Promega), and siRNAs were transfected using Lipofectamine RNAiMAX (Thermo Fisher).

**Gel electrophoresis and IP**. Cells and tissues were lysed in cell lysis buffer containing 1% Nonidet P-40, 150 mM NaCl, 50 mM Tris-HCl, 0.1 mM EDTA, proteinase inhibitors, and protein phosphatase inhibitors. Protein lysate was then resolved by sodium dodecyl sulfate-polyacrylamide gel electrophoresis (SDS-PAGE) and transferred to PVDF membranes (Millipore). The membranes were blocked with skimmed milk, incubated first with primary antibodies (1:2000 dilution for HA, Myc, and β-actin; 1:1000 dilution for all other antibodies), and then with peroxidase-conjugated secondary antibodies. Proteins were visualized with ECL chemiluminescent substrate. For phos-tag gel analysis, manganese (II)–phos-tag SDS-PAGE was performed. In brief, the acrylamide-pendant phos-tag ligand (Phos-tag AAL-107, Wako Laboratory Chemicals) was added in the SDS-PAGE to provide a phosphate affinity and cause mobility shifts of phosphorylated proteins. For IP, cell or tissue lysates were incubated with antibody-coated agarose beads at 4 °C for 2 h. The beads were then washed and boiled in SDS loading buffer, and the proteins were detected by Western blotting. Uncropped images of Western blots can be found in the Source Data file.

**Fluorescent imaging and live cell imaging**. Cells were fixed with 4% paraformaldehyde/PBS for 20 min and permeabilized in 0.5% Triton X-100/PBS for 20 min at room temperature. Cells were blocked with 2% bovine serum albumin/phosphate-buffered saline (BSA/PBS), incubated first with primary antibodies (1:500 dilution), and then with Alexa Fluor 594-conjugated secondary antibodies (1:1000 dilution), followed by staining with BODIPY 493/503 and DAPI. Cells were examined with a TCS SP5 confocal microscope (Leica). For live cell imaging showing the lipid droplet dynamics, differentiated C3H/10T1/2 cells were loaded with BODIPY 558/568 C12 fatty acid, and then treated under various conditions. Images were recorded at 5 min intervals for 2 h using a Zeiss LSM 710 Duo NLO multiphoton microscope equipped with a live cell system.

**Lipolysis assay**. For ex vivo lipolysis assay, WAT explants (~3 mm diameter) and isolated brown adipocytes were prepared[54,55]. Briefly, WAT was dissected and cut into ~3 mm diameter pieces. For brown adipocyte isolation, freshly dissected interscapular BAT was cut into ~2 mm diameter pieces, digested in Krebs-Ringer buffer supplemented with 1% fatty acid free BSA, 0.1% HEPES (1 M, pH 7.3), 0.8 mg/mL Type 2 Collagenase (Worthington, LS004176), and 1.2 mM calcium chloride at 37 °C for 50 min. Digested tissue was then filtered through a 100 μm membrane and centrifuged at 300 g for 3 min. The floating adipocyte fraction was collected, washed twice, and used for following studies. Then WAT explants and isolated brown adipocytes were transferred to Krebs-Ringer solution with 3 mM HEPES and 1% FFA-free BSA and incubated at 37 °C under vigorous shaking. Media samples were collected at desired time points. The free glycerol released into the medium was determined using the Free Glycerol Reagent (Sigma-Aldrich, F6428). To determine lipolysis in cultured cells, cells were washed and incubated in phenol red-free medium. The amount of released glycerol was measured as described above.

**GTEx data**. The dataset (dbGaP Accession phs000424.v8.p2) used for the analyses described in this manuscript were obtained at https://www.ncbi.nlm.nih.gov/projects/gap/cgi-bin/study.cgi?study_id=phs000424.v8.p2. Our access to this dataset for General Research Use has been approved by the National Human Genome Research Institute. The GTEx Project was supported by the Common Fund of the Office of the Director of the National Institutes of Health

(commonfund.nih.gov/GTEx). Additional funds were provided by the NCI, NHGRI, NHLBI, NIDA, NIMH, and NINDS. Donors were enrolled at Biospecimen Source Sites funded by NCI\Leidos Biomedical Research, Inc. subcontracts to the National Disease Research Interchange (10XS170), Roswell Park Cancer Institute (10XS171), and Science Care Inc. (X10S172). The Laboratory, Data Analysis, and Coordinating Center (LDACC) was funded through a contract (HHSN268201000029C) to the The Broad Institute, Inc. Biorepository operations were funded through a Leidos Biomedical Research, Inc. subcontract to Van Andel Research Institute (10ST1035). Additional data repository and project management were provided by Leidos Biomedical Research Inc. (HHSN261200800001E). The Brain Bank was supported supplements to University of Miami grant DA006227. Statistical Methods development grants were made to the University of Geneva (MH090941 & MH101814), the University of Chicago (MH090951, MH090937, MH101825, and MH101820), the University of North Carolina—Chapel Hill (MH090936), North Carolina State University (MH101819), Harvard University (MH090948), Stanford University (MH101782), Washington University (MH101810), and to the University of Pennsylvania (MH101822).

**Statistics**. Statistical tests used are stated in figure legends. Data were plotted and subjected to statistical analysis using GraphPad Prism version 7.0a for Mac OS X (GraphPad Software). Results were presented as mean ± s.e.m. as indicated in Figure legends. Unpaired two-tailed Student's $t$ test was used for two-condition comparison. Two-way analysis of variance (ANOVA) with post hoc Dunnett's test was used for multiple-conditions comparison. Linear regression analyses were performed to determine correlation between two variables. Measurements were taken from distinct samples except tracking experiments in which measurements were also taken at various time points. In some experiments, same groups of mice were used to determine body weight, fat mass, and lean mass.

**Reporting summary**. Further information on research design is available in the Nature Research Reporting Summary linked to this article.

## Data availability

The source data underlying Figs. 1c–e, h–j, l, m, 2a, b, d, f, g, 3b, e, 4a–c, e–k, 5a, d–g, 6a–f, h, k, l, 7b–f, h–k, m, 8a–f and Supplementary Figs. 1a–d, 2a, b, d, 3a, b, d, 4a, b, d, 5a, b, d, f, h, i, 6a–p, 7a–e, 8a–j, 9a, c–g, 10a–d, f, h, 11a–d are provided as a Source Data file. Other datasets generated and/or analyzed in the current study are available from the corresponding author upon reasonable request.

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

## Acknowledgements

We thank Dr. Gerald I. Shulman and Dr. Rachel J. Perry from Yale University, Alexander Munk from University of Copenhagen for insightful advice. This work was supported by the National Institutes of Health (R01DK089098 (X.Y.) and R01DK102648 (X.Y.)) and American Heart Association award #17POST33670873 (Y.Y.).

## Author contributions

Y.Y. performed most of the experiments, analyzed the data, prepared figures and schemes, and wrote the draft of the manuscript. M.F., M.-D.L., K.Z., B.Z., S.W., Y.L. and W.N. helped with conducting some of the biochemistry and animal experiments and data analysis. Q.O. and J.M. helped with the analyses of datasets from GTEx and GeneNetwork. X.Y. and Y.Y. are responsible for the experimental design, data analysis and interpretation, writing, revising, and finalization of the manuscript.

## Competing interests

The authors declare no competing interests.
