## [Peer Review File · Nature Communications]

Reviewers' comments:

Reviewer #1 (Remarks to the Author):

The original paper from Yunfang Yang and collaborators describes the huge work done by the team led by Xiaoyong Yang (Yale University School of Medicine, New Haven) regarding the implication of the glycosyltransferase OGT in the complex process of obesity.

The major claims of the paper are as follows. By using an inducible deletion system the authors studied the function of OGT in the adipose tissue: the enzyme inhibits lipolysis in visceral fat. They largely depicted the molecular mechanisms underlying this important findings; OGT seems to decrease the phosphorylation of the lipid droplet-associated protein PLIN1 that acts as an activator of lipolysis, therefore deleting OGT induces visceral adipose tissue lipolysis and inhibiting the enzyme should contribute to fight obesity. In the other hand when OGT is overexpressed, adipose tissue lipolysis is blocked, obesity and insulin resistance occur.

According to me, the data are novel, the conclusions drawn convincing and the paper should be of interest for the large scientific community.

Nevertheless, I think the quality of the manuscript can be even more increased by responding to the following queries.

At the beginning of the results session, a scheme depicting the localization of the different adipose tissues (BAT, iWAT and eWAT) used in the study (in mice and why not also in humans) would help the comprehending.

The authors say page 5 that "Interestingly, the respiratory exchange ratio (RER; V_{CO2}/V_{O2}) of OGT AKO mice was significantly lower than that of WT controls during the light cycle (Fig. 1c-e), suggesting that OGT ablation promotes whole body utilization of lipids over carbohydrates": what are the arguments supporting this claim ?

It is mainly question in this paper of lipolysis: did the author look at the expression of beta-oxidation enzymes and lipogenesis ones (ACC, FAS, TGS) ?

Generally I have no questioning regarding the experiments that are rather well done and well led (this included statistical interpretations). Nevertheless for some experiments (e.g. those presented in fig 4 and S5) some controls are missing to assert that OGT or OGA overexpression was efficient (anti-O-GlcNAc staining would help).

Another point, the authors used forskolin to stimulate lipolysis (through activation of cAMP synthesis). But it is also known that GFAT, the rate-limiting enzyme of UDP-GlcNAc (the substrate of OGT) synthesis is also sensitive to forskolin. How do the authors discriminate both effects ?

Also HSL phosphorylation is monitored in some experiments but pan-HSL is lacking.

Figure 4 needs to be reordered, panel j being presented before panel i, and while both being linked.

To be complete and also gain clarity, the article is very dense in terms of experiences, a final general scheme explaining the main observations would be appreciable for the reader.

Tony Lefebvre

Reviewer #2 (Remarks to the Author):

In this study, Yang and coworkers show that inducible deletion of adipose OGT (O-GlcNAc transferase) causes a visceral fat loss by promoting lipolysis in visceral fat. At the mechanistic level, visceral fat maintains high levels of O-GlcNAcylation during fasting. Loss of OGT decreases O-GlcNAcylation of PLIN1, which leads to elevated PLIN1 phosphorylation and enhanced lipolysis. Moreover, adipose OGT overexpression inhibits lipolysis and promotes diet-induced obesity. These findings reveal an essential role for OGT in the regulation of adipose tissue homeostasis and suggest that targeting O-GlcNAc signaling in adipose tissue could be of interest for the treatment of obesity.

Major points

1-The authors show a clear link between OGT and lipolysis in eWAT. Since OGT/O-GlcNAcylation was mostly associated with glucose sensing in the past, it would be of interest to determine whether lipogenic gene expression is altered in response to refeeding in eWAT from wt and OGT AKO mice (Fig.2).

2- The authors clearly show that Loss of OGT promotes fasting-induced PLIN1 phosphorylation in visceral fat. Is this observation specific to PLIN1? What about other proteins known to be differentially regulated by phosphorylation and O-GlcNAcylation, such as protein kinase B/Akt. Is fed and/or fasted phosphorylation of Akt modulated in visceral fat from OGT KO mice?

3-Given the phenotype observed, it would be of interest to challenge OGT AKO mice with high fat diet and determine the effect on insulin sensitivity/resistance, hepatic steatosis.

4-The phenotype associated with OGT overexpression in adipose tissue is interesting. O-GlcNAcylation and phosphorylation levels of PLIN1 should be shown and correlated to the phenotype observed (lipolysis). The increased in fat mass of these mice under HFD diet is quite impressive. Could this increased in fat mass also be due to increased lipogenic gene expression?

Reviewer #3 (Remarks to the Author):

The paper by Yang et al. discusses the role of O-GlcNAc transferase (OGT) in adipose tissue and shows that deletion of OGT results in stimulated lipolysis in visceral fat and consequent loss of fat mass partly via O-GlcNAcylation of PLIN1 resulting in phosphorylation of PLIN1. Conversely, overexpressing OGT resulted in inhibition of lipolysis and diet induced obesity.

This study has been performed by using in vivo models (inducible OGT knock out mice and mice overexpressing OGT) along with in vitro studies. A major shortcoming of the study is the lack on any human data, which clearly limits the interpretation and extrapolation of the paper. Other than that, the studies appear well designed, the experiments are carefully conducted and the (overwhelming amount of) data are presented in a clear and concise manner.

Major concerns

The effects of OGT on PLIN1 phosphorylation are novel, unexpected and intriguing. Additional studies to show release of CGI-58 from the lipid droplet and binding of CGI-58 to ATGL to examine if the observed stimulated lipolysis upon deletion of OGT is dependent on ATGL mediated lipolysis are strongly appreciated.

The paper really needs addition of human data to underpin statements like 'These findings indicate a unique potential role of O-GlcNAc signaling in the treatment of obesity' (abstract) or 'data reveal an essential role for OGT in lipolysis regulation and indicate a unique potential for targeting signaling to combat metabolically unhealthy obesity' (end of the intro) as well as the final phrase of the discussion. As the paper is now, we are at risk of just studying a mice phenotype, which may not have any role in human (patho)physiology.

At multiple spots in the paper, conclusions are not underpinned by data (it even says 'data not shown'). I feel this insufficiently matches the need for more openness in science and would recommend including these data of removing the discussion on the data that have not been shown.

Minor issues

I would suggest omitting panel b and panel c from Supplemental figure S2 and have them replace by a panel showing RER values over light and dark periods and bar graphs showing substrate oxidation, that would be more informative and also fit better with the figure caption

Point-by-point Response to Reviewers' Comments

Reviewer #1:

“..... According to me, the data are novel, the conclusions drawn convincing and the paper should be of interest for the large scientific community. Nevertheless, I think the quality of the manuscript can be even more increased by responding to the following queries.”

Thank you for finding this paper novel, convincing and broadly interesting.

1, At the beginning of the results session, a scheme depicting the localization of the different adipose tissues (BAT, iWAT and eWAT) used in the study (in mice and why not also in humans) would help the comprehending.

Thank you for the suggestion. We have now added a scheme (Fig. 1b) showing the localization of various mouse adipose tissues including interscapular brown adipose tissue (BAT), subcutaneous inguinal white adipose tissue (iWAT) and visceral epididymal white adipose tissue (eWAT). The localization of similar types of human adipose tissues is shown alongside.

2, The authors say page 5 that “Interestingly, the respiratory exchange ratio (RER; VCO_2/VO_2) of OGT AKO mice was significantly lower than that of WT controls during the light cycle (Fig. 1c-e), suggesting that OGT ablation promotes whole body utilization of lipids over carbohydrates”: what are the arguments supporting this claim?

The respiratory exchange ratio (RER; VCO_2/VO_2) provides insight into the relative contribution of lipids and carbohydrates to overall energy expenditure. By calculation, a predominant lipid oxidation brings RER down to 0.70, while a predominant carbohydrate oxidation elevates RER up to 1.00. Protein oxidation is usually negligible. RER is widely used as an indicator of fuel selection (lipids vs. carbohydrate) in animals¹. In our study, we observed a decreased RER (closer to 0.7) in OGT AKO mice during light time (Fig. 1e), which indicates a preferred lipid utilization in OGT AKO mice when they are not actively eating (Supplementary Fig. S2b).

3, It is mainly question in this paper of lipolysis: did the author look at the expression of beta-oxidation enzymes and lipogenesis ones (ACC, FAS, TGS)?

This is a good point. As the reviewer requested, we examined the expression of genes involved in beta-oxidation and lipogenesis by quantitative real time PCR (Supplementary Fig. S6). The results showed that the mRNA level of VLCAD, an enzyme essential for fatty acid oxidation, was not affected by OGT knockout. Moreover, the mRNA level of PGC-1 α , a master regulator of mitochondrial biogenesis and fatty acid oxidation, was comparable between WT and OGT KO adipose tissue. The mRNA levels of enzymes mediating triglyceride synthesis, DGAT1 and DGAT2, were not changed in OGT KO adipose tissue. Interestingly, we did observe significant increases in the mRNA levels of ACC1 and FASN in OGT KO adipose tissues under fed and/or fasted conditions, but not during refeeding, suggesting that loss of adipose OGT may promote *de novo* lipogenesis (Supplementary Fig. S6m-o). Whether this is a feedback response to increased lipolysis or independent of OGT-mediated regulation of lipolysis warrants further investigation.

4, Generally I have no questioning regarding the experiments that are rather well done and well led (this included statistical interpretations). Nevertheless, for some experiments (e.g. those presented in fig 4 and S5) some controls are missing to assert that OGT or OGA overexpression was efficient (anti-O-GlcNAc staining would help).

Thank you for these excellent suggestions. O-GlcNAc (RL2) blots for samples used in Fig. 4a, 4g and Supplementary Fig. S8c (now Supplementary Fig. S8g) have now been added (Supplementary Fig. S9a and S9h). RL2 blots for Fig. 3a, Supplementary Fig. S5a, S5d and S5f (now S5c, S5g, and S5j) have also been added (Supplementary Fig. S5a, S5b, S5f and S5i). Taken together, our results clearly show that OGT knockout largely abolished overall O-

GlcNAcylation in adipose tissue. OGT overexpression in HeLa cells dramatically enhanced overall O-GlcNAcylation level while OGA overexpression significantly suppressed overall O-GlcNAcylation. Inhibiting OGT by using ST045849 (ST04) in *in vitro* cultured adipocytes significantly suppressed overall O-GlcNAcylation, while inhibiting OGA by using TMG in *in vitro* cultured adipocytes greatly enhanced overall O-GlcNAcylation. These results demonstrate that genetic and pharmacological manipulation of OGT or OGA is efficient to alter overall O-GlcNAc modification in our *in vivo* and *in vitro* experiments.

5, Another point, the authors used forskolin to stimulate lipolysis (through activation of cAMP synthesis). But it is also known that GFAT, the rate-limiting enzyme of UDP-GlcNAc (the substrate of OGT) synthesis is also sensitive to forskolin. How do the authors discriminate both effects?

A previous study showed that in 24 hour-glucose starved NRK cells (overall O-GlcNAcylation will be largely depleted at this point), glucose-induced recovery of O-GlcNAcylation was suppressed by forskolin treatment, which is due to the inhibitory effect of forskolin on GFAT, a rate-limiting enzyme of UDP-GlcNAc synthesis². However, in our experiments, the timeline of O-GlcNAc manipulation and forskolin treatment was quite different. We first altered the level of O-GlcNAcylation in cells (usually for a relatively long time: overnight or days), then forskolin (with or without IBMX) was added to stimulate lipolysis (usually for a short time: 5 minutes to 1 hour). As shown by our Western blot analysis, forskolin (with or without IBMX) treatment did slightly decreased overall O-GlcNAcylation level in some cases (in Supplementary Fig. S5i and S8f, but not Fig. 6c and Supplementary Fig. S9a). Nevertheless, we observed that OGT knockout dramatically decreased overall O-GlcNAcylation level (Supplementary Fig. S8f and S9a), while OGT overexpression and OGA inhibitor TMG greatly enhanced overall O-GlcNAcylation level (Fig. 6c and Supplementary Fig. S5i). Therefore, the effect of forskolin on overall O-GlcNAcylation is mild in our experimental settings. We also argue that the difference in lipolysis between control and OGT knockout/overexpression groups can still be attributed to the difference in O-GlcNAcylation since forskolin-mediated GFAT regulation is upstream of O-GlcNAc signaling.

6, Also HSL phosphorylation is monitored in some experiments but pan-HSL is lacking. Figure 4 needs to be reordered, panel j being presented before panel i, and while both being linked.

Thanks for the suggestion! We have now added the Western blotting results of pan-HSL (HSL) in Fig. 4a and Supplementary Fig. S8c (now Supplementary Fig. S8g). Panels in Figure 4 have been reordered as requested.

7, To be complete and also gain clarity, the article is very dense in terms of experiences, a final general scheme explaining the main observations would be appreciable for the reader.

A working model summarizing the role of OGT in lipolysis and obesity has now been added (Fig. 7o).

Reviewer #2:

1, Major points 1-The authors show a clear link between OGT and lipolysis in eWAT. Since OGT/OGlcNAcylation was mostly associated with glucose sensing in the past, it would be of interest to determine whether lipogenic gene expression is altered in response to refeeding in eWAT from wt and OGT AKO mice (Fig.2).

Thank you for this great suggestion. In our study, we observed that *ad libitum*-fed WT and OGT AKO mice have similar body weight and body composition. However, short-term (6 hours) fasting induced a more dramatic decrease in fat mass in OGT AKO mice, as compared with WT controls. We then focused on studying the role of OGT in lipolysis.

During a 6-hour refeeding, WT and OGT AKO mice regained similar visceral fat mass (Fig. 2b), suggesting that OGT does not affect *in vivo* lipid accumulation in adipose tissue. Consistent with this idea, the mRNA levels of key enzymes mediating *de novo* lipogenesis and triglyceride synthesis including ACC1, FASN, DGAT1 and DGAT2, were similar between re-fed WT and OGT AKO mice (Supplementary Fig. S6o and S6p).

It's worth mentioning that we did observe significant increases in the mRNA levels of ACC1 and FASN in OGT KO adipose tissues under fed and/or fasted conditions (Supplementary Fig. S6m and S6n). Whether this is a feedback response to increased lipolysis or independent of OGT-mediated regulation of lipolysis warrants further investigation.

2, The authors clearly show that Loss of OGT promotes fasting-induced PLIN1 phosphorylation in visceral fat. Is this observation specific to PLIN1? What about other proteins known to be differentially regulated by phosphorylation and O-GlcNAcylation, such as protein kinase B/Akt. Is fed and/or fasted phosphorylation of Akt modulated in in visceral fat from OGT KO mice?

Thank you for these important questions. Our results showed that mRNA and protein levels of key lipolytic genes in mouse adipose tissue were not affected by OGT depletion (Fig.4 and Supplementary Fig. S6 and S8). The phosphorylation level of HSL in adipose tissue was similar between WT and OGT AKO mice (Fig.4 and Supplementary Fig. S8). Although we cannot exclude the possibility that other regulators of lipolysis may be regulated by O-GlcNAcylation, our results showed that overexpression of phosphomimetic mutant PLIN1-EE, but not PLIN1 or nonphosphorylatable mutant PLIN1-AA, restored the suppressed IBMX/Fsk-induced lipolysis in OGT-overexpressing cells (Fig. 6f). These results demonstrate that the effect of OGT on lipolysis is largely dependent on its activity on PLIN1 phosphorylation.

We also examined the phosphorylation level of Akt as the reviewer suggested. As shown in Supplementary Fig. S8d and S8e, the level of Akt serine 473 phosphorylation is enhanced in visceral fat of OGT AKO mice at the fed state. However, fasting decreased the level of Akt serine 473 phosphorylation in both WT and OGT KO visceral fat. More importantly, we observed no difference in the level of Akt serine 473 phosphorylation in visceral fat of WT and OGT AKO mice. These results suggest that Akt phosphorylation is not likely to contribute to OGT-mediated regulation of lipolysis.

3, Given the phenotype observed, it would be of interest to challenge OGT AKO mice with high fat diet and determine the effect on insulin sensitivity/resistance, hepatic steatosis.

The goal of this study was to use the inducible adipose-specific OGT knockout (OGT AKO) mice to determine the primary effect of adipose OGT depletion. For this purpose, most of the experiments in this study were performed within 2 weeks after tamoxifen-induced OGT deletion in adipose tissue. Our fasting-refeeding experiment suggested that adipose OGT mainly affects fasting-induced fat mass loss (Fig. 1), Therefore, the role of adipose OGT in lipolysis became the main topic of our study. Our glucose and insulin tolerance tests (GTT and ITT) showed that WT and OGT AKO mice had similar glucose tolerance and insulin sensitivity (Supplementary Fig. S4b-e). We detected a mild increase in hepatic triglyceride levels in OGT AKO mice after 6 hours of fasting. However, fasting-induced hepatic triglyceride accumulation in OGT AKO mice was eliminated after 24 hours of refeeding (Supplementary Fig. S4a).

The reviewer's question actually falls within the scope of our other study which has been recently published in Nature Communications (PMID: 30504766)³. In that study, we used the constitutively expressed Adiponectin-Cre to induce long-term adipose (fat)-specific OGT knockout (OGT FKO). We found that OGT FKO greatly reduced high fat diet (HFD)-induced obesity. HFD-fed OGT FKO mice also had improved systemic glucose metabolism and enhanced insulin sensitivity, compared with HFD-fed WT mice. Consistently, decreased accumulation of triacylglycerides and diacylglycerides was found in the liver of OGT FKO mice, indicating attenuated hepatic steatosis in OGT FKO mice. In a nutshell, we think that short-term

adipose OGT depletion mainly promotes lipolysis without significantly affecting whole-body metabolism, whereas long-term adipose OGT depletion may contribute to improved whole-body metabolism by lowering body fat.

4, The phenotype associated with OGT overexpression in adipose tissue is interesting. OGlcNAcylation and phosphorylation levels of PLIN1 should be shown and correlated to the phenotype observed (lipolysis). The increased in fat mass of these mice under HFD diet is quite impressing. Could this increase in fat mass also due to increased lipogenic gene expression?

The levels of O-GlcNAcylation and phosphorylation of endogenous PLIN1 in visceral fat of WT and OGT knockin mice has been examined as requested. The results show that OGT overexpression enhances PLIN1 O-GlcNAcylation and suppresses PLIN1 phosphorylation (Fig. 7i). We also observed decreased O-GlcNAcylation and increased phosphorylation of endogenous PLIN1 in OGT knockout visceral fat (Fig. 6b).

We appreciate the reviewer finding the rapidly increased fat mass in HFD-fed OGT knockin mice impressing. The expression of lipogenic genes have been examined as the reviewer requested. The results show that the mRNA levels of key enzymes mediating *de novo* lipogenesis and triglyceride synthesis including ACC1, FASN, DGAT1 and DGAT2, in visceral fat were similar between HFD-fed WT and OGT AKI mice (Supplementary Fig. S11c and S11d). These results suggest that OGT is unlikely to play a significant role in lipogenesis.

Reviewer #3:

“..... A major shortcoming of the study is the lack on any human data, which clearly limits the interpretation and extrapolation of the paper.”

Thank you for raising this concern. In our original manuscript, we did present human data from the Genotype-Tissue Expression (GTEx) database showing that the OGT/OGA transcript ratio is significantly higher in visceral fat than subcutaneous fat in both Men and Women (Fig. 4k). We also observed that Women have a lower OGT/OGA ratio than Men for both subcutaneous and visceral fat (Fig. 4k). Moreover, an increase in the OGT/OGA ratio during aging was found, especially in visceral fat from Men (Supplementary Fig. S9f and S9g). These results suggest that increased O-GlcNAcylation may serve as a molecular signature for visceral fat in human and it may contribute to the gender and age differences in adipose tissue homeostasis.

In the revised manuscript, we have obtained additional human data including BMI and diagnosis of diabetes from the dbGAP database. Our analysis shows that the OGT/OGA transcript ratio in subcutaneous fat is positively related with BMI (Fig. 8c). Again, visceral fat generally has a higher OGT/OGA ratio but no significant correlation between the OGT/OGA ratio and BMI was observed (Fig. 8d). Human subjects diagnosed with type 2 diabetes (T2D) have higher OGT/OGA ratios than non-diabetic “normal” controls in both subcutaneous fat ($p < 0.0001$) and visceral fat ($p = 0.09$) (Fig. 8e, f). These results suggest that an increased adipose OGT/OGA ratio is a molecular signature of obesity and diabetes in humans.

To substantiate the results of our experiments using the C57BL/6 mouse line, we have now obtained gene expression data and metabolic phenotyping data of ~40 different mouse strains from the GeneNetwork database. Our analysis shows that despite the marked genetic heterogeneity of these mouse strains, fasted serum free fatty acid (FFA) levels in HFD-fed mice are negatively correlated to the OGT/OGA transcript ratio in white fat. A clear trend of negative correlation was also observed in normal chow (NC)-fed mice (Fig. 8a). Moreover, the trends of positive correlation between adipose OGT/OGA ratio and glycemia in glucose tolerance test (glucose area under the curve) were observed in NC and HFD-fed mice ($p = 0.12$ for HFD-fed group) (Fig. 8b). These results suggest a role for adipose O-GlcNAc signaling in the regulation of lipolysis and whole-body metabolism in various mouse strains.

“Other than that, the studies appear well designed, the experiments are carefully conducted and the (overwhelming amount of) data are presented in a clear and concise manner.”

Thank you for the laudatory comments.

1, Major concerns: The effects of OGT on PLIN1 phosphorylation are novel, unexpected and intriguing. Additional studies to show release of CGI-58 from the lipid droplet and binding of CGI-58 to ATGL to examine if the observed stimulated lipolysis upon deletion of OGT is dependent on ATGL mediated lipolysis are strongly appreciated.

Previous studies have shown that during stimulated lipolysis, phosphorylated PLIN1 will disassociate with CGI-58, which allows CGI-58 to bind and activate ATGL, the rate-limiting enzyme in lipolysis. Consistent with these findings, we now show that OGT knockout promotes the release of CGI-58 from lipid droplets and enhances the interaction between CGI-58 and ATGL upon lipolytic stimulation (Fig. 6j and 6k). Moreover, inhibiting ATGL activity by using Atglistatin abolishes the difference in lipolysis between cultured WT and OGT KO adipocytes (Fig. 6l), demonstrating that the effect of OGT on lipolysis is dependent on ATGL-mediated lipolysis.

2, The paper really needs addition of human data to underpin statements like ‘These findings indicate a unique potential role of O-GlcNAc signaling in the treatment of obesity’ (abstract) or ‘data reveal an essential role for OGT in lipolysis regulation and indicate a unique potential for targeting signaling to combat metabolically unhealthy obesity’ (end of the intro) as well as the final phrase of the discussion. As the paper is now, we are at risk of just studying a mice phenotype, which may not have any role in human (patho)physiology.

As described above, additional human data have been added to the revised manuscript.

3, At multiple spots in the paper, conclusions are not underpinned by data (it even says ‘data not shown’). I feel this insufficiently matches the need for more openness in science and would recommend including these data of removing the discussion on the data that have not been shown.

Thank you for pointing this out. We have now gone through the text and have eliminated or updated all the content of “data not shown”.

The following discussion about the role of OGT in brown adipose tissue “*We also examined ... and the results showed that ..., while leaving the cold-induced thermogenesis largely unaffected (unpublished data).*” has been removed. This does not affect the discussion in the manuscript since another independent study published in *Diabetes* (N. Ohashi *et al.* 2017)⁴ showing similar results were included in the discussion.

In another paragraph, we mentioned that “*Our previous result showed that constitutive adipose-specific Ogt knockout mice are resistant to diet-induced obesity (unpublished data).*”. This work has now been published in *Nature Communications*³. The paper has now been cited in the revised manuscript.

4, Minor issues: I would suggest omitting panel b and panel c from Supplemental figure S2 and have them replace by a panel showing RER values over light and dark periods and bar graphs showing substrate oxidation, that would be more informative and also fit better with the figure caption.

Sorry for the confusion. Graphs and statistics of RER over light and dark periods have been shown in the main figure (Fig. 1e-1g). The purpose of supplemental figure S2 is to show that other metabolic parameters including oxygen consumption rate, food intake and physical activity, were comparable between WT and OGT AKO mice. The figure caption of supplemental figure S2 has now been changed to “OGT AKO mice have normal oxygen consumption rate, food intake and physical activity”.

References:

- 1 Coutts, B. E. & McConnell, T. Encyclopedia of Food Sciences and Nutrition. 2d ed. (Book). *Library Journal* **129**, 38-38 (2004).
- 2 Chang, Q. *et al.* Phosphorylation of human glutamine:fructose-6-phosphate amidotransferase by cAMP-dependent protein kinase at serine 205 blocks the enzyme activity. *J Biol Chem* **275**, 21981-21987, doi:10.1074/jbc.M001049200 (2000).
- 3 Li, M. D. *et al.* Adipocyte OGT governs diet-induced hyperphagia and obesity. *Nat Commun* **9**, 5103, doi:10.1038/s41467-018-07461-x (2018).
- 4 Ohashi, N. *et al.* Pivotal Role of O-GlcNAc Modification in Cold-Induced Thermogenesis by Brown Adipose Tissue Through Mitochondrial Biogenesis. *Diabetes* **66**, 2351-2362, doi:10.2337/db16-1427 (2017).

REVIEWERS' COMMENTS:

Reviewer #1 (Remarks to the Author):

I feel that the authors responded adequately to my queries, additional controls have been included and I really appreciate the schemes in fig. 1B and fig. 7o.

The efforts that have been made by the authors for revising the manuscript are consistent.

I can therefore consider the paper suitable for publication.

Reviewer #2 (Remarks to the Author):

The authors have adequately answered the questions raised.

I am fully satisfied with the answers provided and the manuscript has greatly improved.

Reviewer #3 (Remarks to the Author):

To the authors

The revised version of the paper by Yang and Yang is a much more balanced and improved version of the original submission. All essential points that were raised by me have been addressed appropriately.

Apologies for having missed the GTEx data in the original submission, the addition of the human data in the revised version is highly appreciated. The new studies on phosphorylation of PLIN1 and its dissociation of CGI58 and the inhibition of ATGL by atglistatin are convincing and a valuable addition to the paper.

Thanks for adding data or eliminating statement 'data not shown from the paper. I feel we all benefit from transparency in data presentation. The inclusion of the recently accepted paper by the same group in Nat Comm in the reference list is helpful.